# D-TPT: Dimensional Entropy Maximization for Calibrating Test-Time Prompt Tuning in Vision-Language Models

## Abstract

Test-time adaptation paradigm provides flexibility towards domain shifts by performing immediate adaptation on unlabeled target data from the source model. Vision-Language Models (VLMs) leverage their generalization capabilities for diverse downstream tasks, and test-time prompt tuning has emerged as a prominent solution for adapting VLMs. In this work, we explore contrastive VLMs and identify the modality gap caused by a single dominant feature dimension across modalities. We observe that the dominant dimensions in both text and image modalities exhibit high predictive sensitivity, and that constraining their influence can improve calibration error. Building on this insight, we propose dimensional entropy maximization that regularizes the distribution of textual features toward uniformity to mitigate the dependency of dominant dimensions. Our method alleviates the degradation of calibration performance in test-time prompt tuning, offering a simple yet effective solution to enhance the reliability of VLMs in real-world deployment scenarios.

## 1 Introduction

Foundation models provide generalized performance through massive data (Rombach et al., 2022; Oquab et al., 2023; Kirillov et al., 2023); among them, Vision Language Models (VLMs) such as CLIP (Radford et al., 2021) are applied to diverse downstream tasks (Zhang et al., 2021; Goyal et al., 2023). Based on the observation that the zero-shot performance of VLMs varies considerably depending on prompt configuration, prompt tuning methods have been proposed to optimize the prompt and determine more appropriate ones for the target task (Zhou et al., 2022b;a). Prompt tuning-based approaches are applied across various areas, including continual learning (Wang et al., 2022d;c), test-time adaptation (Niu et al., 2024; Zhang et al., 2024c), and visual recognition tasks (Zhou et al., 2023; He et al., 2023), which utilize pretrained models.

Furthermore, Test-time Prompt Tuning (TPT) extends its applicability to unlabeled settings by adapting prompts online (Shu et al., 2022). Although TPT effectively improves predictive accuracy, it simultaneously induces overconfidence due to its reliance on prediction entropy minimization as the objective function. Recent studies (Zhang et al., 2024b; Han et al., 2025) attempt to mitigate the degradation in model calibration performance caused by entropy minimization and improve practical applicability. Notably, calibration methods in test-time adaptation for VLMs are based on empirical observations. While increasing text feature diversity effectively improves calibration (Yoon et al., 2024; Sharifdeen et al., 2025); however diversity-based approaches suffers from a lack of understanding regarding its underlying mechanism.

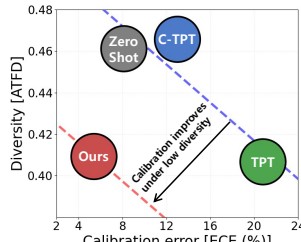

Figure 1: Correlation between text diversity and calibration error.

The correlation between average text feature dispersion (ATFD) and expected calibration error (ECE) (Naeini et al., 2015) has been demonstrated through various experiments (Yoon et al., 2024). Nevertheless, Figure 1 shows that calibration capability can be achieved despite low ATFD, suggesting that underlying causal factors remain to be uncovered. Research aiming to improve calibration

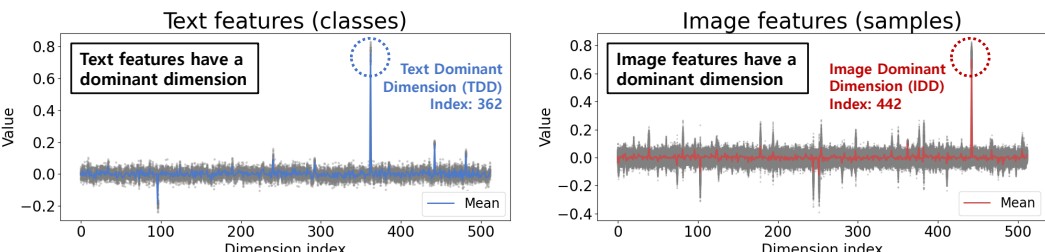

Figure 2: Comparison of feature values across dimensions for text and image features. In CLIP, the modality gap causes features from each modality to be positioned in different spaces, a phenomenon that manifests as the dominant influence of a few dimensions.

capability by regularizing logit norms has achieved promising results in single modal networks (Wei et al., 2022). However empirical evidence shows that logit range plays a more significant role than logit magnitude in CLIP (Murugesan et al., 2024). If text embeddings are interpreted as counterparts to classifier weights in single-modal networks, they naturally exhibit uniformity as a consequence of neural collapse (Papyan et al., 2020). In contrast, VLMs such as CLIP do not follow this structural pattern. Text and image representations in VLM are trained through separate encoders via contrastive learning, exhibiting a modality gap whereby image and text embeddings occupy distinct embedding spaces (Liang et al., 2022). Based on this insight, we conjecture that the modality gap is central to the effectiveness of diversity-based approaches. A detailed discussion of our geometric intuition is provided in Section 4.4.

Contrastive VLMs such as CLIP are trained by maximizing the cosine similarity between image and text embeddings. Nevertheless, a significant modality gap between image and text features is widely observed, and various approaches have been proposed to understand and exploit this phenomenon (Ouali et al., 2023; Oh et al., 2023). **The modality gap is not expressed uniformly across all feature dimensions; instead, it is concentrated in a single dominant dimension.** As depicted in Figure 2, modality gap arises from distinct bases in the image and text representations. These dominant dimensions are consistently observed across all text classes and all image samples. We denote the dimension prevailing in textual embeddings as the text-dominant dimension (TDD) and the corresponding one in visual embeddings as the image-dominant dimension (IDD).

A dominant dimension significantly influences the final logit computed via the inner product of image and text features and exhibits high sensitivity. To investigate the effects of TDD and IDD, we compared predictions in which the TDD and IDD dimensions were replaced with their respective mean values, thereby limiting the contribution of these dominant dimensions. Interestingly, despite restrictions on dominant dimensions, there are cases where both accuracy and ECE are improved. In particular, for TDD, the average ECE is improved for both zero-shot CLIP and TPT. Based on these observations, we propose **d**imensional entropy maximization for **t**est-time **p**rompt **t**uning (D-TPT), a calibration method that mitigates uncertainty by reducing dependence on dominant dimensions and maximizing the entropy of text features to better exploit the remaining dimensions.

We evaluate calibration performance across five calibration metrics on fine-grained classification datasets comprising eleven datasets and natural distribution shifts datasets, which are four ImageNet variants. Our method that regularizes the distribution in intra-text features provides competitive performance compared to existing methods that increase inter-text feature diversity. We note that prior work has shown that the modality gap is primarily induced by a few dominant embedding dimensions (Schrodi et al., 2024). Consistent with this, we independently observed the same phenomenon and confirm that such dominant dimensions are critical to prompt characteristics. Beyond this confirmation, our contribution lies in proposing methods that leverage this hidden property to improve calibration performance.

## 2   RELATED WORK

**Prompt tuning.** Pretrained models are utilized in various applications, and prompt tuning is a parameter-efficient fine-tuning approach that adapts large pretrained models to downstream tasks

by training only a small number of input tokens (Lester et al., 2021). Prompt tuning has become a standard solution alongside LoRA (Hu et al., 2022) and adapters (Rebuffi et al., 2017; Zhang et al., 2021), demonstrating its effectiveness across vision models (Wang et al., 2022d; Jia et al., 2022; Bahng et al., 2022), and language models (Li & Liang, 2021; Liu et al., 2021). In VLMs, CoOp (Zhou et al., 2022b) demonstrates that input contexts such as "a photo of a [class]" significantly influence downstream performance. To overcome the limitations of manual prompt engineering, CoOp introduced context optimization, where prompts are trained rather than hand-crafted. Building on this idea, prompt tuning for VLMs has since been widely explored. Notably, CoCoOp (Zhou et al., 2022a) utilizes image features to generate image-conditioned prompts that enhance generalization to unseen classes, while MaPLe (Khattak et al., 2023) extends context tuning by applying prompts to both the image encoder and the text encoder.

**Test-time adaptation.** Deep neural networks show strong performance in visual tasks but remain vulnerable under distribution shifts (Shimodaira, 2000). Test-time adaptation has been proposed to address dynamically changing real-world environments by enabling pretrained models to adapt during inference in an unsupervised manner, thereby avoiding the need for additional pretraining. Research on test-time adaptation has primarily focused on entropy minimization (Wang et al., 2021; Niu et al., 2023; Lee et al., 2024; Han et al., 2025) and consistency regularization (Wang et al., 2022a; Döbler et al., 2023; Liu et al., 2024). Under this paradigm, TPT (Shu et al., 2022) adopts entropy minimization through augmented input views and prompt tuning, yielding effective adaptation of VLMs during inference and providing a foundation for subsequent research. (Feng et al., 2023; Zhang et al., 2024a; Xiao et al., 2025; Sheng et al., 2025).

**Model calibration.** Modern neural networks often suffer from overconfidence, which leads to miscalibration (Guo et al., 2017). Calibration denotes the alignment between predicted probabilities and the true likelihood of correctness, and it is essential for reliable deployment in real-world settings. Existing methods include post-hoc calibration approaches (Platt et al., 1999; Guo et al., 2017; Zhang et al., 2020) such as temperature scaling and regularization-based approaches (Kumar et al., 2018; Mukhoti et al., 2020) that refine the objective function. Given the calibration problems of test-time adaptation in VLMs, C-TPT (Yoon et al., 2024) improves diversity by encouraging text embeddings to disperse from the centroid. In contrast, SaLS (Murugesan et al., 2024) argues that miscalibration originates from the logit range and proposes a method that adjusts logits within the zero-shot range. Furthermore, O-TPT (Sharifdeen et al., 2025) promotes orthogonality across text features, which enhances class separability and mitigates directional bias.

## 3 PRELIMINARY

**Zero-shot classification.** CLIP (Radford et al., 2021) is pretrained on massive image and text pairs to maximize the cosine similarity between the outputs of the image encoder $E_{visual}$ and the text encoder $E_{text}$. For zero-shot inference, the inputs to the image encoder and the text encoder are an image $x$ and a set of context prompts $\{[\mathrm{p}; y_c]\}_{c=1}^{C}$, where $\mathrm{p}$ denotes the prompt and $y_c$ denotes the class context, where $C$ corresponds to the number of classes. Following previous work (Sharifdeen et al., 2025), we adopt the prompt "a photo of a [class]" as the base prompt. Accordingly, the predicted probability of class $c$ for an image $x_i$ is given by $p_{i,c} = \frac{\exp(\tau \cdot cos(t_c, v_i))}{\sum_{k=1}^{C} \exp(\tau \cdot cos(t_k, v_i))}$, where $v_i = E_{visual}(x_i)$ is the image feature and $t_c = E_{text}([\mathrm{p}; y_c])$ is the text feature, and $\tau$ denotes the learnable logit scaling factor.

**Test-time prompt tuning protocol.** Following test-time prompt tuning approaches (Shu et al., 2022; Yoon et al., 2024; Sharifdeen et al., 2025), we aim to improve performance by updating the prompt $\mathrm{p}$ during inference on a single test sample $\{x_i^{test}, y_i^{test}\} \in \{\mathcal{X}_{test}, \mathcal{Y}_{test}\}$. Here, $\mathcal{Y}_{test}$ is only used for performance evaluation and is not accessed during test-time. During inference, the prompt is tuned in an unsupervised online manner. Specifically, TPT (Shu et al., 2022) generates additional $N-1$ augmented views from a single test image, and optimizes the prompt by minimizing the marginal entropy of the low entropy predictions. The objective function is defined as

$$\mathcal{L}_{\mathrm{TPT}} = H(\bar{p}), \text{ where } \bar{p} = \frac{1}{\mu N} \sum_{i=1}^{N} \mathbb{I}[H(p_i) < \rho] \, p_i. \tag{1}$$

where $H$ denotes the entropy (i.e., $H(p_i) = -\sum_c p_{i,c} \log p_{i,c}$), $\rho$ is a confidence threshold, and $\mu$ denotes cutoff percentile over $N$. Algorithm 1 illustrates the overall procedure of test-time prompt tuning. Based on the initial zero-shot predictions, TPT minimizes the marginal entropy of low entropy predictions, while our proposed D-TPT further improves calibration through the regularization loss function described in Section 4. The prompt is updated via gradient descent on the objective function. After adaptation and prediction for a single test sample, the model is reset to its initial weights.

**Evaluation metrics.** We use five evaluation metrics in this study: four calibration metrics (ECE, AECE, MCE, and AURC) and classification accuracy. Expected calibration error (ECE) (Naeini et al., 2015) measures the average discrepancy between predicted confidence and accuracy, and the adaptive ECE (AECE) (Mukhoti et al., 2020) addresses binning sensitivity by employing adaptive binning. Moreover, the maximum calibration error (MCE) (Naeini et al., 2015) captures the worst-case miscalibration across bins, and the area under the risk–coverage curve (AURC) (Geifman et al., 2018) quantifies how well a model's confidence separates correct from incorrect predictions. For all calibration metrics, lower values indicate better calibration.

---

**Algorithm 1** Pytorch-style pseudo-code of D-TPT

# temperature: $\tau$, threshold: $\rho$
# uniform distribution: U, hyperparameter: $\lambda$
# CLIP procedure
$v$ = image_encoder(image) # $v$ shape: [N,D]
$t$ = text_encoder(text)      # $t$ shape: [C,D]
logits = $\tau *$ cosine_similarity($v, t$)
# TPT procedure
$p$ = softmax(logits)
idx = select_confident_index($p, \rho$)
tpt_loss = average_entropy(p[idx])
# D-TPT procedure
dem_loss = kl_divergence(softmax($t$),U)
total_loss = tpt_loss + $\lambda *$ dem_loss
total_loss.backward()
optimizer.step()

---

# 4 METHOD

## 4.1 DIMENSIONAL SENSITIVITY

Given that CLIP obtains logits based on cosine similarity between image and text features, where the dominant dimension significantly contributes to prediction. In order to quantify the contribution of each feature dimension, we define the sensitivity of individual dimensions as

$$s_i = KLD(p_m||q), \tag{2}$$

where $q = \sigma(\tau \cdot cos(t,v))$ denotes the original prediction probability, $\sigma(\cdot)$ denotes a softmax function, and $p_m$ denotes the prediction obtained by masking the $m$-$th$ dimension, $KLD(\cdot \,||\, \cdot)$ denotes the Kullback–Leibler divergence. Figure 3a presents the sensitivity analysis across feature dimensions, which shows that dominant dimensions exhibit substantially higher sensitivity. Moreover, Figure 3b illustrates performance changes when dominant dimensions are replaced with their mean. For zero-shot CLIP, constraining these dimensions improves accuracy and calibration capability in some cases. For TDD, consistent reductions in average ECE were observed in both zero-shot CLIP and TPT. These results indicate that dominant dimensions have a pronounced impact on prediction, while their sensitivity can increase predictive uncertainty.

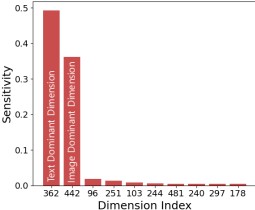
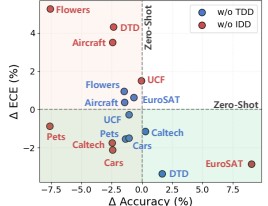

| Dataset | TPT | | w/o TDD | | w/o IDD | |
|---|---|---|---|---|---|---|
| | Acc. (%) ↑ | ECE (%) ↓ | Acc. (%) ↑ | ECE (%) ↓ | Acc. (%) ↑ | ECE (%) ↓ |
| DTD | 46.99 | 20.78 | 47.04 | 17.39 | 45.51 | 27.67 |
| Flowers | 69.06 | 13.29 | 68.86 | 8.91 | 64.11 | 13.56 |
| Aircraft | 23.46 | 16.96 | 22.62 | 17.07 | 21.81 | 27.87 |
| Pets | 87.49 | 5.34 | 87.90 | 4.60 | 87.08 | 6.14 |
| Caltech | 94.32 | 4.70 | 94.12 | 3.57 | 93.96 | 3.27 |
| UCF | 67.94 | 11.72 | 68.20 | 10.22 | 66.93 | 16.88 |
| EuroSAT | 42.62 | 20.50 | 42.68 | 19.46 | 42.52 | 30.71 |
| Cars | 66.20 | 5.43 | 66.12 | 5.49 | 65.07 | 9.64 |
| Mean | 62.26 | 12.34 | 62.19 | 10.84 | 60.87 | 16.97 |

(a) Sensitivity analysis.   (b) Effect of dominant dimensions.

Figure 3: Analysis of the impact of dominant dimensions. **(a)** We present the top-10 values and their indices for sensitivity, which is the change in the prediction distribution when masking the values of each dimension. For both modalities, the dominant dimensions TDD and IDD significantly influence predictions. **(b)** Accuracy and ECE are reported with TDD and IDD replaced by their class-wise mean values. For zero-shot CLIP (left), replacing TDD leads to average improvements in ECE across datasets. For TPT (right), replacing TDD also yields a consistent reduction in mean ECE.

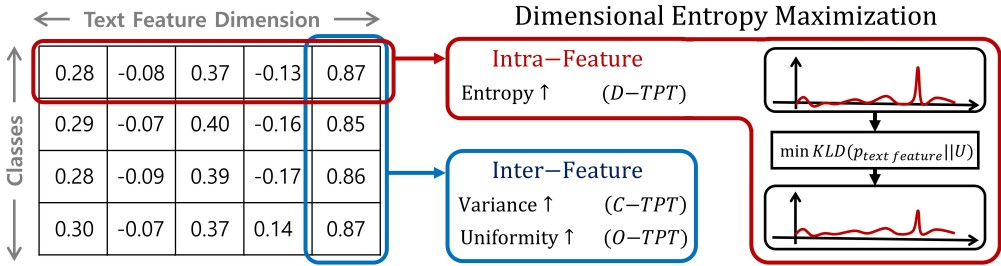

Figure 4: Conceptual illustration of D-TPT. Compared to inter-feature diversity based methods, we achieve improved calibration capability by regularization the distribution of intra-class text features.

## 4.2 D-TPT: DIMENSIONAL ENTROPY MAXIMIZATION FOR TEST-TIME PROMPT TUNING

Building upon an analysis of dominant dimensionality and sensitivity, we introduce dimensional entropy maximization (DEM). We regularize the text feature distribution across all dimensions to be close to a uniform distribution, to reduce the effect of dominant dimensions and increase the contribution of hidden dimensions. Accordingly, our regularization loss term $\mathcal{L}_{DEM}$ is defined as

$$\mathcal{L}_{DEM} = \frac{1}{C} \sum_{c=1}^{C} KLD(\sigma(\bar{t}_c) \| U), \tag{3}$$

where $U$ denotes the uniform distribution and $\bar{t}_c$ denotes the normalized text feature (i.e., $\bar{t}_c = \frac{t_c}{\|t_c\|_2}$) corresponding to the $c$-th class. Using $\mathcal{L}_{DEM}$ as a regularization term for calibration, it can be easily applied to TPT via a plug-and-play approach. Hence, the total loss function of the proposed D-TPT is as follows:

$$\mathcal{L}_{D-TPT} = \mathcal{L}_{TPT} + \lambda \cdot \mathcal{L}_{DEM}, \tag{4}$$

where $\lambda$ is a hyperparameter. The prompts are updated by minimizing the overall loss function.

## 4.3 COMPARISON WITH C-TPT AND O-TPT

C-TPT (Yoon et al., 2024) and O-TPT (Sharifdeen et al., 2025) are calibration-focused variants of TPT (Shu et al., 2022) that aim to enhance the diversity of text features. Specifically, C-TPT induces text features to move away from the centroid, which increases the variance between text embedding,

$$\mathcal{L}_{C-TPT} = \mathcal{L}_{TPT} - \lambda \cdot \frac{1}{C} \sum_{c=1}^{C} \left\| \bar{t}_c - \frac{1}{C} \sum_{j=1}^{C} \bar{t}_j \right\|_2. \tag{5}$$

Similarly, O-TPT maximizes the orthogonality between text features by minimizing the pairwise cosine similarity. Formally, the objective function of O-TPT is formulated as

$$\mathcal{L}_{O-TPT} = \mathcal{L}_{TPT} + \lambda \cdot \|TT^\top - I_C\|_2^2, \tag{6}$$

where $T$ denotes the matrix of normalized text features, and $I_C$ is the $C$-dimensional identity matrix. Figure 4 illustrates the comparison between our D-TPT and existing methods. The primary distinction is the direction of regularization. Previous methods focus on inter-feature structure across classes, while we focus on the intra-feature distribution. Despite this difference, both approaches improve calibration capability in practice. This difference naturally raises the question of why regularization strategies for text features contribute to improved calibration. While a complete theoretical framework remains an open challenge, we present a geometric analysis of the modality gap that provides a principled step toward addressing this question.

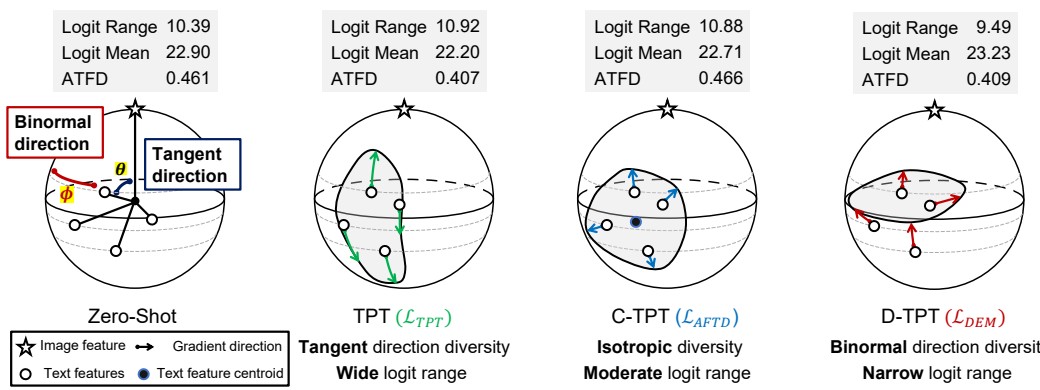

Figure 5: Geometric interpretation of prompt tuning methods on the hypersphere. On the hypersphere, the *tangent direction* represents the geodesic from a text feature toward its corresponding image feature, the *normal direction* corresponds to the radial axis from the center of the hypersphere to the feature, and the *binormal direction* is defined as orthogonal to both tangent and normal directions. In this context, TPT aligns text features along the tangent direction of the hypersphere, which expands the logit range and leads to overconfidence. Meanwhile, C-TPT encourages outward displacement from the text centroid, thereby improving isotropic diversity. In contrast, proposed D-TPT maximizes entropy across feature dimensions, amplifying binormal components and reducing the modality gap.

## 4.4 GEOMETRIC SUPPORT

Figure 5 presents a geometric interpretation of the shift of text features under TPT, C-TPT, and proposed D-TPT. In the single image case, the gradient of the TPT objective function is given by

$$\frac{\partial \mathcal{L}_{tpt}}{\partial \theta} = -\sum_{i=1}^{C} \left(\log p_{i,c} + 1\right) p_{i,c}(1 - p_{i,c})\frac{\partial z_{i,c}}{\partial \theta}, \tag{7}$$

where $z_{i,c}$ and $p_{i,c}$ denote logit and probability distribution corresponding to class $c$ for input $x_i$, respectively, and $\theta$ denotes the model parameters. In TPT, the adoption of a confidence threshold ensures high-confidence prediction distributions. The prompt is optimized to increase the probability of the top-1 class while decreasing the probabilities assigned to the remaining classes. Since CLIP normalizes all feature vectors to unit length, the representations are constrained on a hypersphere. In this geometric structure, TPT encourages the text feature most similar to the image to move closer along the tangent direction of the hypersphere, while pushing other text features farther away. Consequently, this geometric effect enlarges the logit range and induces overconfidence.

In contrast, C-TPT optimizes text features to move away from the centroid. Let $k$ denote the number of text features that are closer to the image feature than the centroid, C-TPT increases the cosine similarity between the image feature and the top-$k$ text features. This mechanism mitigates overconfidence by compensating for the similarity reduction among non-top-1 classes. However, text features located farther from the centroid are explicitly assigned lower probabilities, which inherently carries the risk of increasing the logit range.

Our proposed D-TPT reduces the influence of dominant bases in the image features, thereby narrowing the modality gap. By maximizing entropy across all feature dimensions, D-TPT amplifies the binormal directions on the hypersphere. Empirically, our method exhibits similar ATFD to TPT but with a lower logit range, suggesting diversity in binormal directions. Since the logit mean is proportional to the average cosine similarity between image and text feature, we can confirm the reduction of the modality gap. Therefore, the reduced logit range of D-TPT leads to a decrease in softmax probabilities, which mitigates overconfidence.

**From dominant dimension to overconfidence.** We present the empirical results of the logit range, logit mean, and ATFD on the DTD in Figure 5. For CLIP-ViT-B/16, the logit mean is 22.90. Given a logit-scaling factor of $\tau = 100$ and using the relation $z_{i,c} = \tau \cos(t_c, v_i)$, the average angle between image and text features becomes $\arccos\left(\frac{22.90}{100}\right) = 76.8°$. This indicates that the image and text representations are separated by a modality gap close to 90°. The dominant dimensions in the image

| Fine-Grained Classification | | | | |
|---|---|---|---|---|
| **Method** | CLIP-ViT-B/16 | | | |
| | Acc. (%) ↑ | ECE (%) ↓ | AECE (%) ↓ | MCE (%) ↓ | AURC ($\times 10^3$) ↓ |
| Zero Shot | 63.84 | 4.25 | 4.15 | 20.03 | 184.28 |
| TPT | $65.09_{\pm0.16}$ | $11.42_{\pm0.19}$ | $11.37_{\pm0.20}$ | $31.62_{\pm5.02}$ | $185.64_{\pm0.68}$ |
| C-TPT | $64.46_{\pm0.15}$ | $4.97_{\pm0.19}$ | $5.11_{\pm0.16}$ | $21.54_{\pm3.44}$ | $187.50_{\pm0.46}$ |
| O-TPT | $63.98_{\pm0.13}$ | $4.78_{\pm0.13}$ | $4.88_{\pm0.13}$ | $20.47_{\pm5.55}$ | $187.25_{\pm0.13}$ |
| D-TPT (*Ours*) | $64.72_{\pm0.07}$ | $4.18_{\pm0.16}$ | $4.18_{\pm0.19}$ | $21.18_{\pm7.27}$ | $181.67_{\pm0.30}$ |
| | CLIP-RN50 | | | |
| Zero Shot | 56.05 | 5.25 | 5.29 | 17.37 | 254.15 |
| TPT | $58.07_{\pm0.19}$ | $11.27_{\pm0.18}$ | $11.26_{\pm0.17}$ | $27.72_{\pm3.87}$ | $251.15_{\pm1.25}$ |
| C-TPT | $57.57_{\pm0.12}$ | $6.20_{\pm0.15}$ | $6.16_{\pm0.18}$ | $23.96_{\pm5.06}$ | $243.87_{\pm0.50}$ |
| O-TPT | $57.36_{\pm0.10}$ | $5.63_{\pm0.16}$ | $5.68_{\pm0.13}$ | $20.78_{\pm3.78}$ | $244.96_{\pm0.45}$ |
| D-TPT (*Ours*) | $57.13_{\pm0.12}$ | $5.91_{\pm0.14}$ | $5.90_{\pm0.13}$ | $22.44_{\pm5.30}$ | $245.27_{\pm1.00}$ |

Table 1: Calibration performance comparison between D-TPT and other baselines using CLIP-ViT-B/16 (above) and CLIP-RN50 (below) backbones on fine-grained classification datasets.

and text features serve as different basis directions, and their orthogonality leads to a modality gap that is nearly orthogonal.

The analysis in prior work (Murugesan et al., 2024) demonstrates that expanding the logit range amplifies softmax confidence and results in overconfidence. In CLIP, logits are computed from the cosine similarity between image and text features; therefore, the logit range corresponds to the angular difference between the maximum and minimum image–text pairs on the hypersphere. An increase in angular dispersion along the tangent direction directly widens the logit range, which in turn induces overconfidence.

Both the image and text features have dominant dimensions that impose a consistent directional tendency within the hyperspherical feature space. Reducing the dominant dimension of the text features shifts them toward the image feature direction. Since feature vectors are normalized to unit length, the reduced energy in the dominant dimension is redistributed across the remaining dimensions. Specifically, the logit ranges for TPT, C-TPT, and D-TPT are 10.92, 10.88, and 9.49, respectively. For ATFD, the values are 0.407 for TPT, 0.466 for C-TPT, and 0.409 for D-TPT. Although D-TPT exhibits a similar level of text dispersion to TPT, its smaller logit range demonstrates that D-TPT achieves diversity primarily along the binormal direction, rather than expanding excessively along the tangent direction. Consequently, the proposed method improves calibration by narrowing the tangent direction range and enhancing discriminability along the binormal direction.

## 5 EXPERIMENTS

### 5.1 EXPERIMENTAL SETUP

**Datasets.** Following the standard evaluation protocol for downstream tasks in VLMs (Zhou et al., 2022b; Yoon et al., 2024), we use datasets under natural domain shift, including ImageNet-A (Hendrycks et al., 2021b), ImageNet-V2 (Recht et al., 2019), ImageNet-R (Hendrycks et al., 2021a), and ImageNet-Sketch (Wang et al., 2019). For fine-grained classification, we use ImageNet (Deng et al., 2009), DTD (Cimpoi et al., 2014), Flowers102 (Nilsback & Zisserman, 2008), Food101 (Bossard et al., 2014), SUN397 (Xiao et al., 2010), FGVC-Aircraft (Maji et al., 2013), Oxford Pets (Parkhi et al., 2012), Caltech101 (Li et al., 2022), UCF101 (Soomro et al., 2012), EuroSAT (Helber et al., 2018), and Stanford Cars (Krause et al., 2013). Experimental results are reported as the average performance across all datasets for both the natural domain shift and fine-grained classification datasets. Moreover, we present the mean and standard deviation over three trials with random seeds 0, 1, and 2.

**Implementation details.** We adopt TPT as the baseline and reproduce results under identical settings to enable further analysis of calibration metrics. Following TPT, each test sample is evaluated with a

| | Natural Distribution Shifts | | | | |
|---|---|---|---|---|---|
| | CLIP-ViT-B/16 | | | | |
| Method | Acc. (%) ↑ | ECE (%) ↓ | AECE (%) ↓ | MCE (%) ↓ | AURC ($\times 10^3$) ↓ |
| Zero Shot | 57.19 | 4.93 | 4.90 | 10.54 | 216.23 |
| TPT | $60.24_{\pm 0.10}$ | $11.77_{\pm 0.12}$ | $11.74_{\pm 0.10}$ | $21.72_{\pm 1.95}$ | $211.24_{\pm 0.62}$ |
| C-TPT | $58.33_{\pm 0.15}$ | $5.42_{\pm 0.07}$ | $5.45_{\pm 0.07}$ | $12.11_{\pm 1.07}$ | $210.52_{\pm 0.41}$ |
| O-TPT | $58.48_{\pm 0.07}$ | $5.14_{\pm 0.07}$ | $5.17_{\pm 0.14}$ | $11.33_{\pm 0.82}$ | $209.91_{\pm 0.15}$ |
| D-TPT (*Ours*) | $57.87_{\pm 0.07}$ | $3.83_{\pm 0.04}$ | $3.85_{\pm 0.07}$ | $9.59_{\pm 1.26}$ | $211.31_{\pm 0.24}$ |
| | CLIP-RN50 | | | | |
| Zero Shot | 40.66 | 7.19 | 7.13 | 18.66 | 395.77 |
| TPT | $43.35_{\pm 0.09}$ | $17.06_{\pm 0.14}$ | $17.05_{\pm 0.15}$ | $32.21_{\pm 1.27}$ | $383.07_{\pm 0.84}$ |
| C-TPT | $41.76_{\pm 0.07}$ | $8.99_{\pm 0.12}$ | $8.97_{\pm 0.10}$ | $21.52_{\pm 0.53}$ | $390.19_{\pm 0.48}$ |
| O-TPT | $41.74_{\pm 0.09}$ | $8.97_{\pm 0.13}$ | $9.00_{\pm 0.14}$ | $21.36_{\pm 1.79}$ | $390.24_{\pm 0.80}$ |
| D-TPT (*Ours*) | $42.92_{\pm 0.06}$ | $8.05_{\pm 0.09}$ | $8.07_{\pm 0.07}$ | $20.43_{\pm 1.01}$ | $379.73_{\pm 0.38}$ |

Table 2: Calibration performance comparison between D-TPT and other baselines using CLIP-ViT-B/16 (above) and CLIP-RN50 (below) backbones on natural distribution shifts datasets.

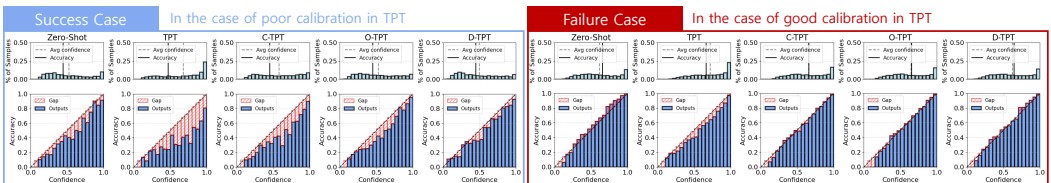

Figure 6: Confidence histograms and reliability diagrams. We present the sample distribution across confidence levels, as well as the gap between accuracy and confidence, for a successful case (DTD) and a failure case (Stanford Cars).

batch size of 64, comprising the original image and 63 augmented images. We use CLIP-ViT-B/16 and CLIP-RN50 as backbone architectures. Prompts are initialized with "a photo of a [class]" and optimized using AdamW (Loshchilov & Hutter, 2017) with a learning rate of 0.005. All experiments are conducted on NVIDIA RTX A6000 GPUs with 48 GB of memory. For practical applicability in test-time settings, we fix the hyperparameter $\lambda = 10^5$ across all datasets and architectures.

## 5.2 EXPERIMENTAL RESULTS

**Fine-grained classification.** Table 1 shows the average performance across eleven fine-grained datasets for evaluation on downstream tasks. Compared with zero-shot CLIP, TPT consistently shows improved accuracy but worse calibration performance. Designed for improved calibration, C-TPT, O-TPT, and D-TPT apply additional regularization for calibration. These calibration-oriented methods achieve improved accuracy than zero-shot CLIP while reducing calibration error compare to TPT. Among them, D-TPT achieves 0.74%, 0.60%, and $5.58 \times 10^3$ improvements in accuracy, ECE, and AURC, respectively, compared to the previous state-of-the-art O-TPT on CLIP-ViT-B/16. On the other hand, O-TPT demonstrates the best performance in MCE, indicating the need for calibration evaluation across the various metrics. We do not achieve the best performance on CLIP-RN50, however, we achieve competitive performance between C-TPT and O-TPT. Detailed results are provided in A.1.

**Natural distribution shifts.** Table 2 shows the average performance across four ImageNet variants, consisting of ImageNet-A/R/V2/Sketch for out-of-distribution evaluation. For CLIP-ViT-B/16, we achieve 0.61% lower performance and 1.31% improved calibration error compared to the previous state-of-the-art O-TPT. Compared to the zero-shot CLIP, we achieve improved accuracy and ECE by 0.68% and 1.10%, respectively, demonstrating the potential for improvement across all metrics through test-time prompt tuning. Furthermore, for CLIP-RN50, D-TPT demonstrates improved performance across all metrics compared to existing methods. Detailed results are provided in A.2.

## 5.3 FURTHER ANALYSIS

**Failure case analysis.** Figure 6 presents confidence histograms (above) and reliability diagrams (below) to analyze the distribution of predictions and calibration trends with respect to confidence. Following C-TPT (Yoon et al., 2024), we report average accuracy and confidence over 20 bins. From both success and failure cases in comparison with existing methods, we observe that D-TPT more closely preserves the confidence distribution of the zero-shot CLIP. Therefore, the beneficial effects of D-TPT are particularly pronounced in cases where TPT significantly increases overconfidence. Figure 7 shows the ECE of C-TPT and D-TPT across high and low ECE intervals of TPT, confirming that D-TPT is particularly effective in high-risk settings where TPT tends to amplify overconfidence.

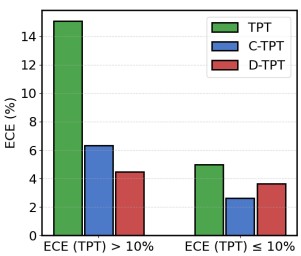

Figure 7: C-TPT vs D-TPT.

Furthermore, this observation is supported by the geometric interpretation discussed in Section 4.4. Specifically, D-TPT aims to align all class text features with the image feature, hence mitigating changes in the features of non-top-1 classes induced by TPT. Whereas C-TPT alleviates such changes primarily for classes that are closer to the centroid than the top-1 prediction. Consequently, our method can be understood as particularly effective in scenarios where TPT amplifies calibration error, since it counteracts the update directions for a broader set of class text features.

**Pareto front analysis.** Regularization methods for calibration improve calibration capability but often suffer from accuracy degradation (Kumar et al., 2018; Karandikar et al., 2021; Yoon et al., 2024). Regarding the trade-off between accuracy and calibration performance, Figure 8 presents the variations in performance of C-TPT, O-TPT, and D-TPT according to the coefficient of the regularization term. Increasing marker size indicates a larger influence of the regularization loss with increasing hyperparameter $\lambda$. Experimental results demonstrate that increasing $\lambda$ leads to a reduction in accuracy but a steady improvement in ECE. Notably, all methods achieve improvements in ECE without degradation in accuracy on Flowers102, indicating that these calibration approaches offer an enhanced solution rather than a simple compromise between accuracy and calibration.

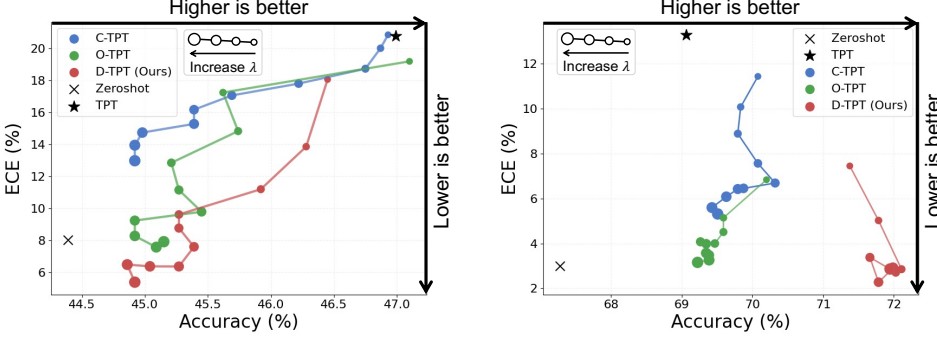

Figure 8: Pareto front analysis. We show the variation in ECE and Accuracy on DTD (left) and Flowers102 (right) as the hyperparameter $\lambda$ increases. Regarding the trade-off between Accuracy and ECE, a point closer to the right-under area indicates better performance.

**Effect of prompt initialization.** We present the performance of different prompt initialization schemes in Table 3. As described in the implementation details, we use the hand-crafted prompt "a photo of a [class]" as the initial prompt. Additionally, we conduct experiments using a prompt trained from CoOp (Zhou et al., 2022b) and another hand-crafted prompt, "a picture of a [class]", to assess robustness to prompt initialization. In both cases, our method consistently improves ECE. Especially, under the initialization "a picture of a [class]", our method achieves the best performance on most evaluation metrics.

| Fine-Grained Classification | | | | | | | | | |
|---|---|---|---|---|---|---|---|---|---|
| Method | Trained prompt (CoOp (Zhou et al., 2022b)) | | | | | Hand-crafted prompt ("a picture of a") | | | | |
| | Acc. ↑ | ECE ↓ | AECE ↓ | MCE ↓ | AURC ↓ | Acc. ↑ | ECE ↓ | AECE ↓ | MCE ↓ | AURC ↓ |
| CLIP-ViT-B/16 | 63.11 | 5.20 | 5.27 | 28.78 | 189.92 | 64.50 | 4.11 | 4.16 | 32.76 | 178.13 |
| TPT | 63.96 | 18.58 | 18.51 | 42.44 | 190.82 | 65.68 | 10.86 | 10.84 | 31.89 | 180.00 |
| C-TPT | 64.04 | 9.95 | 9.90 | 29.51 | 188.13 | 65.23 | 5.11 | 5.19 | 23.32 | 177.90 |
| O-TPT | 63.60 | 7.80 | 7.73 | 27.97 | 187.97 | 64.59 | 4.51 | 4.50 | 32.43 | 179.07 |
| D-TPT (*Ours*) | 63.20 | 6.96 | 6.96 | 24.43 | 195.41 | 65.22 | 4.31 | 4.37 | 24.28 | 171.27 |

Table 3: Effect of initial prompt setting. We report accuracy (%), ECE (%), AECE (%), MCE (%) and AURC ($\times 10^3$) for the trained prompt by CoOp and the hand-crafted prompt "a picture of a [class]".

## 6    CONCLUSION AND LIMITATIONS

In this paper, we introduce D-TPT, a simple yet effective solution to improve the calibration capability of test-time prompt tuning. Based on observations of dominant feature dimensions in contrastive VLMs, we demonstrate that constraining dominant features to specific dimensions can be beneficial in mitigating overconfidence. In contrast to diversity-based methods, our observations imply that useful hidden factors remain beyond diversity for improving the calibration of VLMs. Furthermore, we provide a geometric perspective on the modality gap to explain the success of regularization-based approaches for text features. Empirical results on standard VLM benchmarks confirm that D-TPT improves calibration performance.

**Limitation and future work.** Nonetheless, our geometric interpretation is grounded in the dynamics of text features for a single image feature, leaving a practical gap in assumptions compared to approaches utilizing multiple image features via augmentation. Future work will aim to close the theoretical gap regarding the dynamics of text features and improvements in calibration performance, building a theoretical foundation based on the modality gap.

**Policy on large language model**: We used ChatGPT for minor english editing and language polishing of the manuscript.

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

# A APPENDIX

## A.1 DETAILED RESULTS ON FINE-GRAINED CLASSIFICATION

Table 4 and 5 show the calibration performance for fine-grained classification across individual datasets. We report performance averaged over three different random seeds.

## A.2 DETAILED RESULTS ON NATURAL DISTRIBUTION SHIFTS DATASETS

Table 6 and 7 present the dataset-wise average performance of CLIP-ViT-B/16 and CLIP-RN50 under natural distribution shifts, respectively. Our code and logs are provided in the supplementary materials.

| Method | ImageNet | DTD | Flowers102 | Food101 | SUN397 | FGVC-Aircraft | OxfordPets | Caltech101 | UCF101 | EuroSAT | StanfordCars | Mean |
|---|---|---|---|---|---|---|---|---|---|---|---|---|
| | | | | | Accuracy (%) ↑ | | | | | | | |
| Zero-Shot | 66.72 | 44.33 | 67.19 | 83.66 | 62.53 | 23.73 | 88.06 | 93.23 | 65.16 | 42.06 | 65.58 | 63.84 |
| TPT | 68.93 | 46.87 | 68.72 | 84.69 | 65.47 | 23.07 | 87.02 | 94.13 | 68.19 | 42.67 | 66.28 | 65.09 |
| C-TPT | 68.10 | 45.31 | 69.24 | 83.09 | 64.50 | 23.93 | 88.28 | 93.41 | 65.03 | 42.41 | 65.70 | 64.46 |
| O-TPT | 67.30 | 44.86 | 69.08 | 82.76 | 63.26 | 23.42 | 88.19 | 93.25 | 63.83 | 42.60 | 65.21 | 63.98 |
| D-TPT | 67.39 | 44.96 | 71.54 | 83.11 | 64.59 | 23.94 | 88.29 | 93.09 | 66.71 | 44.02 | 64.25 | 64.72 |
| | | | | | ECE (%) ↓ | | | | | | | |
| Zero-Shot | 1.91 | 8.09 | 3.19 | 2.06 | 2.10 | 5.51 | 4.34 | 5.02 | 2.93 | 7.13 | 4.50 | 4.25 |
| TPT | 10.53 | 21.25 | 13.55 | 4.33 | 11.25 | 17.16 | 5.81 | 4.51 | 11.49 | 20.37 | 5.33 | 11.42 |
| C-TPT | 3.10 | 12.75 | 5.48 | 3.34 | 5.02 | 4.39 | 1.81 | 3.90 | 2.25 | 11.22 | 1.46 | 4.97 |
| O-TPT | 1.94 | 8.14 | 3.56 | 4.15 | 8.45 | 3.85 | 2.41 | 3.95 | 2.90 | 11.43 | 1.80 | 4.78 |
| D-TPT | 2.73 | 5.20 | 2.82 | 4.07 | 4.50 | 3.74 | 2.38 | 5.43 | 2.65 | 9.78 | 2.62 | 4.18 |
| | | | | | AECE (%) ↓ | | | | | | | |
| Zero-Shot | 1.90 | 7.82 | 2.55 | 2.01 | 2.02 | 5.74 | 4.38 | 5.01 | 2.79 | 7.02 | 4.44 | 4.15 |
| TPT | 10.50 | 21.11 | 13.55 | 4.23 | 11.20 | 17.16 | 5.79 | 4.40 | 11.46 | 20.37 | 5.35 | 11.37 |
| C-TPT | 3.04 | 12.89 | 5.74 | 3.46 | 5.07 | 4.90 | 1.86 | 3.96 | 2.45 | 11.29 | 1.53 | 5.11 |
| O-TPT | 1.91 | 8.12 | 3.88 | 4.35 | 8.45 | 4.50 | 2.36 | 4.04 | 2.67 | 11.44 | 2.00 | 4.88 |
| D-TPT | 2.73 | 5.21 | 2.98 | 4.28 | 4.50 | 3.96 | 2.25 | 5.45 | 2.56 | 9.63 | 2.42 | 4.18 |
| | | | | | MCE (%) ↓ | | | | | | | |
| Zero-Shot | 6.26 | 19.24 | 11.48 | 8.68 | 5.51 | 21.14 | 23.96 | 90.67 | 8.96 | 14.34 | 10.12 | 20.03 |
| TPT | 19.05 | 40.79 | 29.93 | 8.71 | 19.88 | 45.69 | 45.62 | 74.13 | 23.88 | 29.71 | 10.47 | 31.62 |
| C-TPT | 7.98 | 26.61 | 16.75 | 8.88 | 12.56 | 14.61 | 27.69 | 72.82 | 18.21 | 22.39 | 8.44 | 21.54 |
| O-TPT | 11.10 | 22.79 | 15.27 | 9.77 | 29.33 | 13.35 | 18.55 | 58.70 | 10.37 | 23.33 | 12.63 | 20.47 |
| D-TPT | 7.20 | 18.94 | 13.82 | 9.54 | 11.66 | 15.89 | 55.83 | 60.22 | 12.40 | 18.99 | 8.55 | 21.18 |
| | | | | | AURC ($\times 10^3$) ↓ | | | | | | | |
| Zero-Shot | 138.50 | 313.46 | 112.32 | 54.83 | 182.54 | 577.84 | 21.07 | 12.37 | 122.47 | 355.43 | 136.23 | 184.28 |
| TPT | 139.80 | 314.63 | 108.63 | 57.10 | 179.19 | 580.73 | 27.02 | 14.78 | 108.84 | 376.11 | 135.25 | 185.64 |
| C-TPT | 132.15 | 326.02 | 99.24 | 55.37 | 191.14 | 563.65 | 19.89 | 10.76 | 130.99 | 393.63 | 139.63 | 187.50 |
| O-TPT | 132.89 | 313.74 | 98.17 | 55.39 | 182.02 | 570.29 | 19.92 | 10.92 | 137.56 | 396.89 | 141.94 | 187.25 |
| D-TPT | 135.37 | 317.81 | 93.06 | 51.91 | 164.52 | 563.30 | 19.82 | 11.75 | 114.45 | 378.85 | 147.57 | 181.67 |

Table 4: Full results on fine-grained classification using the CLIP-ViT-B/16 backbone.

# B EXPERIMENTAL RESULTS ON MEDICAL DOMAINS

We evaluate the generalization capability of our method in the medical domain, a setting where calibration is particularly important. To this end, we incorporate C-TPT, O-TPT, and D-TPT into BAPLe (Hanif et al., 2024) by adding each corresponding loss function to the original objective

| Method | ImageNet | DTD | Flowers102 | Food101 | SUN397 | FGVC-Aircraft | OxfordPets | Caltech101 | UCF101 | EuroSAT | StanfordCars | Mean |
|---|---|---|---|---|---|---|---|---|---|---|---|---|
| | | | | | Accuracy (%) ↑ | | | | | | | |
| Zero-Shot | 58.13 | 40.37 | 61.59 | 73.93 | 58.73 | 15.81 | 83.59 | 86.21 | 58.84 | 23.70 | 55.60 | 56.05 |
| TPT | 60.74 | 41.86 | 62.47 | 74.98 | 61.43 | 17.57 | 84.58 | 87.41 | 60.79 | 28.51 | 58.42 | 58.07 |
| C-TPT | 60.05 | 41.78 | 64.70 | 74.81 | 60.95 | 16.60 | 83.43 | 87.17 | 60.26 | 27.24 | 56.32 | 57.57 |
| O-TPT | 58.99 | 41.84 | 65.57 | 74.62 | 59.62 | 17.06 | 83.17 | 86.77 | 59.93 | 27.66 | 55.70 | 57.36 |
| D-TPT | 60.39 | 42.40 | 60.29 | 74.24 | 59.80 | 16.93 | 83.13 | 86.99 | 58.72 | 30.30 | 55.27 | 57.13 |
| | | | | | ECE (%) ↓ | | | | | | | |
| Zero-Shot | 1.94 | 8.66 | 2.99 | 2.68 | 3.73 | 6.26 | 5.55 | 4.06 | 2.93 | 14.67 | 4.27 | 5.25 |
| TPT | 11.38 | 25.40 | 13.82 | 5.17 | 9.03 | 15.64 | 3.80 | 4.20 | 10.87 | 20.83 | 3.84 | 11.27 |
| C-TPT | 3.02 | 21.08 | 4.23 | 1.72 | 3.19 | 11.27 | 2.92 | 2.73 | 3.05 | 13.32 | 1.64 | 6.20 |
| O-TPT | 3.13 | 16.72 | 1.78 | 1.38 | 6.58 | 8.25 | 3.14 | 3.01 | 2.47 | 13.51 | 1.99 | 5.63 |
| D-TPT | 2.68 | 13.53 | 4.72 | 5.10 | 4.02 | 7.79 | 5.29 | 5.25 | 2.44 | 6.14 | 8.04 | 5.91 |
| | | | | | AECE (%) ↓ | | | | | | | |
| Zero-Shot | 1.85 | 8.79 | 3.49 | 2.62 | 3.77 | 5.93 | 5.54 | 4.28 | 3.26 | 14.51 | 4.14 | 5.29 |
| TPT | 11.35 | 25.37 | 13.82 | 5.15 | 9.00 | 15.61 | 3.93 | 4.01 | 10.81 | 20.82 | 3.93 | 11.26 |
| C-TPT | 3.03 | 21.03 | 4.29 | 1.75 | 3.16 | 11.27 | 2.77 | 2.66 | 3.06 | 13.32 | 1.46 | 6.16 |
| O-TPT | 3.10 | 16.62 | 3.21 | 1.22 | 6.59 | 8.08 | 3.11 | 2.91 | 2.05 | 13.50 | 2.08 | 5.68 |
| D-TPT | 2.69 | 13.60 | 4.82 | 5.21 | 4.04 | 7.74 | 5.15 | 5.28 | 2.36 | 5.96 | 8.01 | 5.90 |
| | | | | | MCE (%) ↓ | | | | | | | |
| Zero-Shot | 4.33 | 20.31 | 9.15 | 8.16 | 10.83 | 20.30 | 11.37 | 27.51 | 22.36 | 46.19 | 10.61 | 17.37 |
| TPT | 18.97 | 45.88 | 30.73 | 10.06 | 17.26 | 49.33 | 24.14 | 16.40 | 20.57 | 62.62 | 8.96 | 27.72 |
| C-TPT | 7.99 | 43.47 | 13.06 | 8.09 | 20.54 | 36.68 | 20.28 | 35.73 | 10.67 | 58.58 | 8.52 | 23.96 |
| O-TPT | 10.45 | 34.87 | 11.78 | 6.15 | 10.56 | 31.18 | 23.67 | 33.15 | 7.41 | 53.95 | 5.39 | 20.78 |
| D-TPT | 6.21 | 31.62 | 13.07 | 9.15 | 16.17 | 35.65 | 20.54 | 31.93 | 8.24 | 59.77 | 14.49 | 22.44 |
| | | | | | AURC ($\times 10^3$) ↓ | | | | | | | |
| Zero-Shot | 202.07 | 374.38 | 153.86 | 94.24 | 206.22 | 690.02 | 35.37 | 29.24 | 166.81 | 634.18 | 209.32 | 254.15 |
| TPT | 195.57 | 371.36 | 160.31 | 94.53 | 202.81 | 667.79 | 34.31 | 36.17 | 170.28 | 631.39 | 198.15 | 251.15 |
| C-TPT | 192.87 | 363.66 | 135.67 | 90.08 | 198.30 | 677.65 | 36.89 | 27.69 | 159.00 | 591.52 | 209.21 | 243.87 |
| O-TPT | 196.42 | 362.02 | 130.74 | 90.84 | 199.86 | 674.67 | 38.73 | 28.42 | 163.93 | 594.92 | 213.97 | 244.96 |
| D-TPT | 194.21 | 372.90 | 161.13 | 93.94 | 210.55 | 675.49 | 38.76 | 30.45 | 176.89 | 528.28 | 215.39 | 245.27 |

Table 5: Full results on fine-grained classification using the CLIP-RN50 backbone.

function. Since BAPLe does not adopt the episodic inference strategy of TPT, we scale $\lambda$ by $10^{-5}$, $10^{-4}$, and $10^{-4}$ for C-TPT, O-TPT, and D-TPT, respectively. Experiments are conducted on the COVID (Rahman et al., 2021), RSNA18 (Shih et al., 2019), and KatherColon (Kather et al., 2019) datasets, using PLIP (Huang et al., 2023) as the baseline model for KatherColon and MedCLIP (Wang et al., 2022b) for COVID and RSNA18. As shown in Table 8, we achieve competitive calibration performance, demonstrating its applicability to medical domains.

## C  ADDITIONAL ANALYSIS

Figure 9 presents a dimensional analysis of image and text embeddings across various datasets and architectures. Across the five datasets, a dominant dimension emerges consistently for both CLIP-ViT-B/16 and CLIP-RN50, and this dominant dimension appears in the same dimension regardless of the dataset for each backbone. Furthermore, Figure 10 shows sensitivity analysis results obtained across various datasets and backbones. While some variations exist across datasets, these results confirm that the dominant dimension of the TDD consistently exhibits high sensitivity. Tables 9 and 10 present the zero-shot and TPT performance on CLIP-RN50 when the TDD and IDD dimensions are replaced with their mean values, respectively. Consistent with the trends observed in CLIP-ViT-B/16, constraining the influence of the TDD in CLIP-RN50 also leads to a reduction in average ECE, and

| Method | ImageNet-A | ImageNet-V2 | ImageNet-R | ImageNet-Sketch | Mean |
|---|---|---|---|---|---|
| Accuracy (%) ↑ | | | | | |
| Zero-Shot | 47.80 | 60.85 | 73.98 | 46.12 | 57.19 |
| TPT | 53.15 | 62.90 | 76.95 | 47.97 | 60.24 |
| C-TPT | 49.41 | 61.75 | 74.85 | 47.31 | 58.33 |
| O-TPT | 49.91 | 61.67 | 75.22 | 47.11 | 58.48 |
| D-TPT | 49.12 | 61.35 | 74.05 | 46.97 | 57.87 |
| ECE (%) ↓ | | | | | |
| Zero-Shot | 8.43 | 2.89 | 3.55 | 4.87 | 4.93 |
| TPT | 16.03 | 11.63 | 4.46 | 14.98 | 11.77 |
| C-TPT | 7.04 | 4.50 | 2.85 | 7.30 | 5.42 |
| O-TPT | 7.33 | 4.27 | 1.99 | 6.96 | 5.14 |
| D-TPT | 6.35 | 2.66 | 3.85 | 2.47 | 3.83 |
| AECE (%) ↓ | | | | | |
| Zero-Shot | 8.40 | 2.74 | 3.59 | 4.87 | 4.90 |
| TPT | 15.94 | 11.63 | 4.43 | 14.97 | 11.74 |
| C-TPT | 7.15 | 4.49 | 2.87 | 7.30 | 5.45 |
| O-TPT | 7.43 | 4.22 | 2.05 | 6.96 | 5.17 |
| D-TPT | 6.38 | 2.70 | 3.88 | 2.45 | 3.85 |
| MCE (%) ↓ | | | | | |
| Zero-Shot | 17.76 | 7.85 | 8.79 | 7.75 | 10.54 |
| TPT | 27.15 | 22.37 | 11.71 | 25.67 | 21.72 |
| C-TPT | 16.99 | 11.32 | 7.22 | 12.89 | 12.11 |
| O-TPT | 16.56 | 10.92 | 5.81 | 12.01 | 11.33 |
| D-TPT | 16.95 | 7.86 | 8.26 | 5.27 | 9.59 |
| AURC ($\times 10^3$) ↓ | | | | | |
| Zero-Shot | 8.40 | 2.74 | 3.59 | 4.87 | 4.90 |
| TPT | 15.94 | 11.63 | 4.43 | 14.97 | 11.74 |
| C-TPT | 7.15 | 4.49 | 2.87 | 7.30 | 5.45 |
| O-TPT | 7.43 | 4.22 | 2.05 | 6.96 | 5.17 |
| D-TPT | 6.38 | 2.70 | 3.88 | 2.45 | 3.85 |

Table 6: Full results on natural distribution shifts datasets using the CLIP-ViT-B/16 backbone.

we additionally observe that suppressing the IDD further decreases ECE. These additional analyses across diverse datasets and architectures demonstrate that our observations regarding the modality gap constitute a consistent and recurrent phenomenon.

# D COMBINATION WITH C-TPT

Table 11 presents the results of C-TPT, D-TPT, and their combination methods across 10 fine-grained datasets. Each loss function is added to the TPT objective without any hyperparameter adjustment, using 50 for C-TPT and $10^5$ for D-TPT, and all experiments are conducted with the CLIP-ViT-B/16 backbone. Although incorporating C-TPT into D-TPT improves ECE for DTD, Flowers102, FGVC-Aircraft, and StanfordCars, we also observe cases with marginal or degradation, and accuracy consistently decreases. Motivated by failure cases analysis, we report results for D-TPT×C-TPT, which selectively applies C-TPT or D-TPT based on entropy. D-TPT×C-TPT achieves ECE improvements of 1.05% and 0.10% compared to D-TPT and C-TPT, respectively, while attaining a mean accuracy of 64.44%, with performance comparable to D-TPT and 0.34% higher than C-TPT.

| Method | ImageNet-A | ImageNet-V2 | ImageNet-R | ImageNet-Sketch | Mean |
|--------|-----------|-------------|------------|-----------------|------|
| Accuracy (%) ↑ | | | | | |
| Zero-Shot | 21.80 | 51.30 | 56.17 | 33.35 | 40.66 |
| TPT | 24.90 | 54.18 | 59.20 | 35.14 | 43.35 |
| C-TPT | 22.35 | 53.44 | 56.95 | 34.29 | 41.76 |
| O-TPT | 22.72 | 52.99 | 57.26 | 33.98 | 41.74 |
| D-TPT | 24.01 | 53.78 | 59.21 | 34.69 | 42.92 |
| ECE (%) ↓ | | | | | |
| Zero-Shot | 21.24 | 3.38 | 0.96 | 3.19 | 7.19 |
| TPT | 31.50 | 13.55 | 9.35 | 13.86 | 17.06 |
| C-TPT | 22.77 | 5.15 | 1.42 | 6.60 | 8.99 |
| O-TPT | 24.30 | 3.64 | 2.68 | 5.25 | 8.97 |
| D-TPT | 19.93 | 4.12 | 3.66 | 4.50 | 8.05 |
| AECE (%) ↓ | | | | | |
| Zero-Shot | 21.24 | 3.07 | 1.03 | 3.19 | 7.13 |
| TPT | 31.50 | 13.53 | 9.32 | 13.86 | 17.05 |
| C-TPT | 22.77 | 5.13 | 1.39 | 6.60 | 8.97 |
| O-TPT | 24.30 | 3.81 | 2.64 | 5.25 | 9.00 |
| D-TPT | 19.91 | 4.22 | 3.66 | 4.50 | 8.07 |
| MCE (%) ↓ | | | | | |
| Zero-Shot | 53.79 | 9.22 | 4.61 | 7.03 | 18.66 |
| TPT | 59.11 | 25.02 | 18.65 | 26.06 | 32.21 |
| C-TPT | 54.57 | 12.96 | 4.25 | 14.31 | 21.52 |
| O-TPT | 54.80 | 10.16 | 8.20 | 12.26 | 21.36 |
| D-TPT | 50.48 | 11.47 | 7.37 | 12.41 | 20.43 |
| AURC ($\times 10^3$) ↓ | | | | | |
| Zero-Shot | 695.94 | 259.96 | 188.28 | 438.88 | 395.77 |
| TPT | 672.47 | 251.35 | 179.22 | 429.27 | 383.07 |
| C-TPT | 690.08 | 248.86 | 188.44 | 433.38 | 390.19 |
| O-TPT | 687.95 | 248.85 | 188.70 | 435.47 | 390.24 |
| D-TPT | 667.63 | 248.11 | 173.44 | 429.73 | 379.73 |

Table 7: Full results on natural distribution shifts datasets using the CLIP-RN50 backbone.

| | | Clean | | | | Backdoor | | |
|--------|-------|--------|--------|--------|-------|--------|--------|--------|
| Method | BAPLe | w/ C-TPT | w/ O-TPT | w/ D-TPT | BAPLe | w/ C-TPT | w/ O-TPT | w/ D-TPT |
| Baseline: MedCLIP, Dataset: COVID | | | | | | | | |
| Acc (%) | 81.65 | 82.45 | 82.30 | 81.80 | 98.83 | 99.95 | 99.95 | 99.75 |
| ECE (%) | 20.33 | 22.91 | 15.60 | 12.85 | 20.99 | 26.01 | 5.70 | 4.79 |
| Baseline: MedCLIP, Dataset: RSNA18 | | | | | | | | |
| Acc (%) | 60.10 | 47.23 | 60.70 | 60.77 | 98.83 | 99.97 | 99.43 | 99.40 |
| ECE (%) | 15.74 | 18.53 | 15.28 | 17.09 | 10.20 | 13.45 | 11.65 | 11.45 |
| Baseline: PLIP, Dataset: KasherColon | | | | | | | | |
| Acc (%) | 88.80 | 88.79 | 88.82 | 88.80 | 98.28 | 98.27 | 98.32 | 98.29 |
| ECE (%) | 2.54 | 2.53 | 2.56 | 2.51 | 1.46 | 1.46 | 1.46 | 1.50 |

Table 8: Results on medical domains with C-TPT, O-TPT, and D-TPT applied to BAPLe

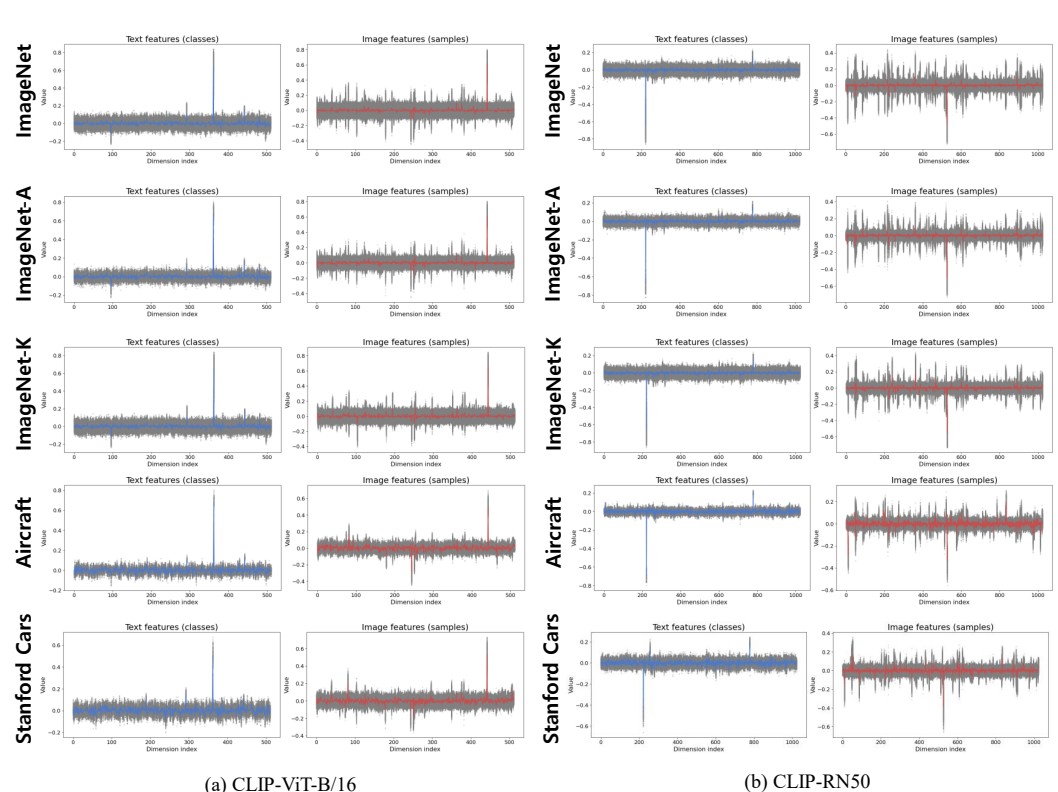

(a) CLIP-ViT-B/16                    (b) CLIP-RN50

Figure 9: Effects of dominant dimensions across datasets and backbones.

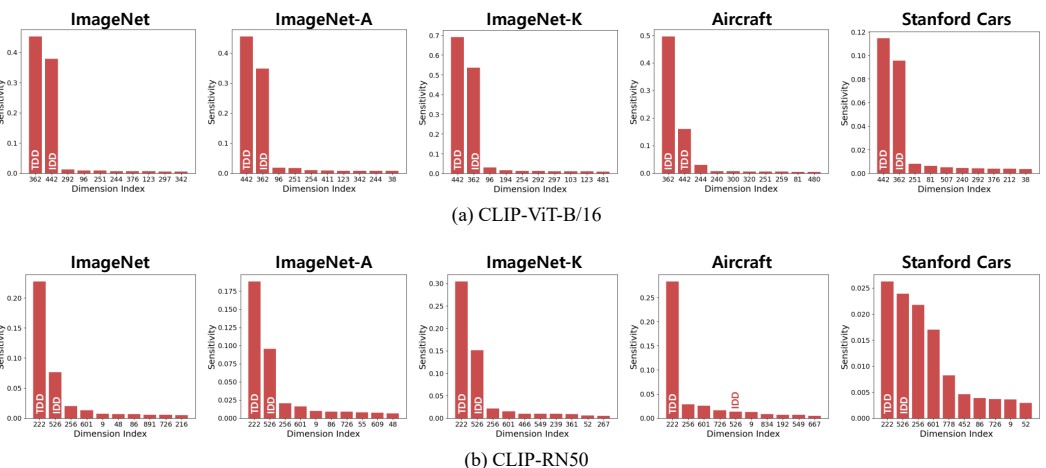

(a) CLIP-ViT-B/16

(b) CLIP-RN50

Figure 10: Sensitivity analysis across datasets and backbones.

| Dataset | Zero Shot | | w/o TDD | | w/o IDD | |
|---|---|---|---|---|---|---|
| | Acc. (%) ↑ | ECE (%) ↓ | Acc. (%) ↑ | ECE (%) ↓ | Acc. (%) ↑ | ECE (%) ↓ |
| DTD | 40.48 | 9.00 | 39.36 | 9.47 | 38.59 | 11.75 |
| Flowers102 | 61.71 | 3.06 | 60.90 | 3.14 | 59.44 | 3.50 |
| FGVC-Aircraft | 15.69 | 6.35 | 15.30 | 6.52 | 15.78 | 5.96 |
| OxfordPets | 83.59 | 5.65 | 82.04 | 4.66 | 84.03 | 5.96 |
| Caltech101 | 85.80 | 3.87 | 85.76 | 3.20 | 87.30 | 4.61 |
| UCF101 | 58.84 | 3.09 | 59.27 | 3.06 | 59.21 | 3.54 |
| EuroSAT | 23.72 | 14.61 | 23.49 | 15.11 | 23.96 | 13.19 |
| StanfordCars | 55.70 | 4.38 | 56.03 | 4.18 | 55.96 | 4.55 |
| Mean | 53.19 | 6.25 | 52.77 | 6.17 | 53.03 | 6.63 |

Table 9: Effect of dominant dimensions on Zero Shot CLIP-RN50

| Dataset | TPT | | w/o TDD | | w/o IDD | |
|---|---|---|---|---|---|---|
| | Acc. (%) ↑ | ECE (%) ↓ | Acc. (%) ↑ | ECE (%) ↓ | Acc. (%) ↑ | ECE (%) ↓ |
| DTD | 41.43 | 26.13 | 41.43 | 25.19 | 41.31 | 25.73 |
| Flowers102 | 62.44 | 13.81 | 63.09 | 11.78 | 62.40 | 10.86 |
| FGVC-Aircraft | 17.73 | 15.33 | 17.73 | 15.06 | 17.61 | 14.78 |
| OxfordPets | 84.49 | 3.88 | 84.74 | 2.89 | 84.98 | 2.69 |
| Caltech101 | 87.10 | 3.68 | 86.90 | 3.73 | 87.51 | 4.34 |
| UCF101 | 60.98 | 10.76 | 61.01 | 10.24 | 60.77 | 10.57 |
| EuroSAT | 28.44 | 21.00 | 28.37 | 21.11 | 29.64 | 14.80 |
| StanfordCars | 58.26 | 3.79 | 58.36 | 3.62 | 58.48 | 3.56 |
| Mean | 55.11 | 12.30 | 55.20 | 11.70 | 55.34 | 10.92 |

Table 10: Effect of dominant dimensions on CLIP-RN50 with TPT

| Method | DTD | Flowers102 | Food101 | SUN397 | FGVC-Aircraft | OxfordPets | Caltech101 | UCF101 | EuroSAT | StanfordCars | Mean |
|---|---|---|---|---|---|---|---|---|---|---|---|
| Accuracy (%) ↑ | | | | | | | | | | | |
| C-TPT | 45.51 | 69.22 | 83.00 | 64.31 | 24.09 | 88.44 | 93.55 | 64.90 | 42.32 | 65.70 | 64.10 |
| D-TPT | 45.09 | 71.78 | 83.10 | 64.46 | 24.06 | 88.25 | 93.14 | 66.67 | 44.09 | 64.21 | 64.49 |
| D-TPT + C-TPT | 44.68 | 71.34 | 83.09 | 64.13 | 23.64 | 88.23 | 92.82 | 65.29 | 40.14 | 64.32 | 63.77 |
| D-TPT × C-TPT | 44.33 | 71.62 | 82.95 | 64.65 | 24.06 | 88.39 | 93.51 | 66.11 | 40.05 | 64.68 | 64.44 |
| ECE (%) ↓ | | | | | | | | | | | |
| C-TPT | 12.54 | 5.48 | 3.25 | 4.89 | 4.29 | 1.65 | 3.96 | 2.85 | 11.49 | 1.19 | 5.16 |
| D-TPT | 5.21 | 2.33 | 4.05 | 4.33 | 3.77 | 2.36 | 5.49 | 2.53 | 9.68 | 2.27 | 4.20 |
| D-TPT+C-TPT | 4.84 | 1.85 | 4.04 | 5.14 | 3.63 | 2.51 | 6.02 | 2.67 | 12.08 | 2.12 | 4.49 |
| D-TPT×C-TPT | 5.32 | 2.82 | 3.59 | 5.07 | 3.80 | 1.98 | 4.17 | 2.28 | 9.70 | 1.83 | 4.10 |

Table 11: Performance comparison of C-TPT, D-TPT, and combined methods.

