# OpenReview forum: "D-TPT: Dimensional Entropy Maximization for Calibrating Test-Time Prompt Tuning in Vision-Language Models"
_ICLR.cc/2026/Conference — Submitted to ICLR 2026_

### Official Review · Reviewer_5wvu · 2025-10-27

**Soundness:** 3
**Presentation:** 4
**Contribution:** 2
**Rating:** 4
**Confidence:** 4

**Summary:**

This paper studies calibration issues in test-time prompt tuning for VLMs. The authors observe that feature distributions in both textual and visual modalities are dominated by a single dimension, causing prediction overconfidence and sensitivity. To mitigate this, the authors introduce Dimensional Entropy Maximization regularization, encouraging uniformity across text embedding dimensions. Extensive experiments across fine-grained datasets and natural domain shift datasets demonstrate that D-TPT improves calibration.

**Strengths:**

1. This paper analyzes the calibration issue from a new perspective, focusing on dominant feature dimension, which is different from prior diversity-based interpretations.
2. The authors provide extensive experiments demonstrating that DTPT shows beneficial effects on calibration under both fine-grained classification and natural distribution changes.
3. The paper is clearly written and well-organized.

**Weaknesses:**

1. The primary motivation relies heavily on analysis from individual examples shown in Fig. 2 and Fig. 3. Can this hold across more samples, more datasets, and different backbone architectures?
2. Although calibration results appear strong, D-TPT does not consistently perform better than C-TPT in both accuracy and calibration.
3. Eq. (1) incorrectly formulates the TPT objective, while TPT actually minimizes the marginal entropy for the mean predictions of the selected augmented views. Due to this incorrect objective, the analysis in Eq. (7) and the related geometric interpretation become questionable.
4. “dominant dimension > modality gap > overconfidence”, this causal chain still lacks deeper formal proof and intuitive explanations.

**Questions:**

1. What is the difference between $\bar{t}_c$ in formula 3 and $t_c$ defined in the PRELIMINARY?
2. Can the proposed method be combined with test-time methods other than TPT, such as TTL?

---

> ### Author Response · Authors · 2025-11-20
>
> We thank the reviewer for the constructive feedback. We address all the concerns and provide additional analysis to support our motivation.
>
> ### **W1. Additional visualization analysis for Figure 2 and Figure 3**
> We include graphs for each dimension across various datasets and backbone architectures in Appendix C. Specifically, Figure 8 of the Appendix C shows that a dominant dimension consistently appears in CLIP-ViT/16 and CLIP-RN across ImageNet, ImageNet-A, ImageNet-K, FGVC-Aircraft, and StanfordCars. Additionally, Figure 9 of the Appendix C presents experimental results on sensitivity. While some variations exist across datasets, these results confirm that the dominant dimension of the text dominant dimension consistently exhibits high sensitivity.
>
> ### **W2. Performance comparison with C-TPT**
> Our method does not always achieve better performance than C-TPT. Nevertheless, we believe it is still meaningful because it shows performance improvement patterns that differ from those of existing diversity-based methods. These patterns suggest that the learning dynamics underlying calibration performance differ from prior approaches, indicating that multiple alternative strategies could be explored to further improve calibration. In particular, we show that D-TPT demonstrates more pronounced improvements than C-TPT when the overconfidence problem caused by TPT is significant. The average ECE for the 7 datasets where TPT's ECE exceeds 10% and the average ECE for the remaining 4 datasets are summarized in the table below.
>
> |ECE (%)|TPT|C-TPT|D-TPT|
> |-|-|-|-|
> |ECE$_{TPT}\>10%$|15.09|6.32|4.49|
> |ECE$_{TPT}\le10%$|5.00|2.63|3.63|
>
> ### **W3. Objective function of TPT**
> Thank you for pointing out the incorrect formula. We corrected the mistake for TPT in Eq (1) in the main text, and you can find it in the revised version. For our implementation, we followed the official TPT code as provided in the supplementary material, and all experimental results were conducted using the marginal entropy minimization method corresponding to the corrected formula. Note that the geometric interpretation of marginal entropy in Figure 5 remains consistent, as it assumes a single image sample to control for prediction changes caused by augmentations.
>
> ### **W4. Causal relation**
> To understand the success of diversity-based approaches, we aimed to leverage CLIP’s modality characteristics and interpret their behavior through the lens of the modality-gap phenomenon. Our central claim is that *diversity is not the everything for calibration*; calibration can still improve even when diversity is low, indicating that factors beyond diversity also contribute to calibration. Through our analysis, we observed that the modality gap manifests primarily through a dominant dimension, and we confirmed that this dimension exhibits high sensitivity with respect to predictions. Although zero-shot CLIP is generally well-calibrated, indicating that the modality gap itself is not the fundamental causal factor behind overconfidence, we argue that understanding and improving calibration nonetheless requires considering the modality gap. As part of this perspective, we propose a simple-yet-effective modality gap based solution and demonstrate that it can improve calibration capability.
>
>
> ### **Q1. The difference between $\bar{t}_c$ and $t_c$**
> As described below Eq (3), $\bar{t}$ denotes the normalized text feature, defined for a text feature t as $\bar{t}=\frac{t}{||t||_2}$. The formula for $\bar{t}$ has been added in the main text.
>
> ### **Q2. Applicability in TTL**
> For C-TPT, O-TPT, and the proposed D-TPT, which are approaches specialized for prompt-tuning–based methods, it is difficult to directly apply them to TTL as claimed in [1] (TTL employs LoRA-based parameter-efficient fine-tuning in LLMs). Nevertheless, within the family of parameter-efficient fine-tuning techniques, prompt tuning remains a major solution alongside LoRA and adapter-based methods. Regarding applications to LLMs, our view is that LLMs not pre-trained with contrastive learning exhibit characteristics fundamentally different from VLMs such as CLIP. Therefore, we conjecture that diversity-based methods may not transfer effectively to LLMs without any revision of the method, although a deeper investigation is needed.
>
> ### **Reference**
> [1] Hu, Jinwu, et al. "Test-Time Learning for Large Language Models.", ICML 2025.

---

> > ### Comment · Reviewer_5wvu · 2025-11-27
> >
> > Thank you for the clarifications and additional analyses.
> >
> > However, my main concern regarding the causal chain “dominant dimension → modality gap → overconfidence” remains unresolved. Considering also that the methodological novelty appears limited, I decide to keep my score.

---

> > > ### Author Response · Authors · 2025-11-28
> > >
> > > We thank the reviewer for the careful reading and comments. Below, we provide further clarification on the causal chains and our novelty.
> > >
> > > ## **1. Dominant dimension $\rightarrow$ Modality gap**
> > > For CLIP-ViT-B/16, the logit mean on the DTD dataset is 22.90. Given a logit-scaling factor of $\tau = 100$, and using the relation
> > >
> > > $\text{logit} = \tau \cos(t, v),$
> > >
> > > the average angle between image and text features is
> > >
> > > $\arccos\left(\frac{22.90}{100}\right) = 76.8^\circ.$
> > >
> > > This indicates that the image and text representations are separated by a modality gap close to $90^\circ$. A dominant dimension in either the image or text space effectively defines a principal direction for that modality, and misalignment between these directions drives the two modalities toward subspaces that are nearly orthogonal. Consequently, a strong dominant dimension induces a modality gap that increases the representational separation between image and text features.
> > >
> > > ## **2. Modality gap $\rightarrow$ Overconfidence**
> > > According to Proposition 2 in [1], an increased logit range amplifies the softmax confidence and consequently leads to overconfidence. In CLIP, logits are computed from the cosine similarity between image and text features; thus, the logit range corresponds to the angular difference between the maximum and minimum image–text pairs on the hypersphere. An increase in angular dispersion (i.e., variance along the geodesic or tangent direction) enlarges this difference, thereby inducing overconfidence.
> > >
> > > |Method|TPT|C-TPT|D-TPT|
> > > |-|:-:|:-:|:-:|
> > > |Logit Range|10.92|10.88|9.49|
> > > |ECE|20.78|12.54|5.21|
> > >
> > > Image embeddings and text embeddings become locally clustered due to the modality gap, and a consistent alignment direction emerges between the two modalities as a result of a dominant dimension. A reduction in the dominant dimension of the text representation shifts the feature vector toward the image feature direction, and because the vector norm is fixed to one, the reduced energy in the dominant dimension is redistributed to the remaining dimensions. This redistribution increases variation along the binormal direction, which enhances discriminability. Furthermore, overconfidence can be reduced by decreasing the range of text features in the tangent direction with respect to image features.  We present both the geometric interpretation in Figure 5 of the main paper and empirical measurements of the logit range, logit mean, and ATFD on the DTD dataset. The logit ranges for TPT, C-TPT, and D-TPT are 10.92, 10.88, and 9.49, respectively. For ATFD, the values are 0.407 for TPT, 0.466 for C-TPT, and 0.409 for D-TPT. Although D-TPT exhibits a similar level of text dispersion to TPT, its smaller logit range demonstrates that D-TPT achieves diversity primarily along the binormal direction, rather than expanding excessively along the tangent direction. Consequently, the proposed method improves calibration by narrowing the tangent-direction range and enhancing discriminability along the binormal direction.
> > >
> > > |Method|TPT|C-TPT|D-TPT|
> > > |-|:-:|:-:|:-:|
> > > |Logit Range|10.92|10.88|9.49|
> > > |ATFD|0.407|0.466|0.409|
> > >
> > > ## **3. Novelty**
> > > In our response to Reviewer zkoL (w1.innovation), we clarified the novelty of our method by contrasting it with zero-directed regularization approaches developed for single-modal settings. Moreover, in our reply to Reviewer oN3E (Official Response by Authors (1/2). Additional explanation on novelty), we summarized the differences between C-TPT, O-TPT, and the proposed D-TPT. In summary, we leverage the different behaviors of single-modal and multimodal representations to deeply understand and improve calibration in VLMs. To the best of our knowledge, this is the first work that (i) interprets diversity-based calibration methods through the lens of the modality gap and (ii) achieves calibration through dimensional entropy maximization.
> > >
> > > ## **Reference**
> > > [1] Murugesan, Balamurali, et al. "Robust calibration of large vision-language adapters.", ECCV 2024.

---

### Official Review · Reviewer_oN3E · 2025-10-27

**Soundness:** 2
**Presentation:** 3
**Contribution:** 2
**Rating:** 4
**Confidence:** 4

**Summary:**

This paper proposes **D-TPT (Dimensional Entropy Maximization for Test-Time Prompt Tuning)** to improve model calibration in vision-language models like CLIP. The key idea is that a few **dominant feature dimensions** cause modality gaps and overconfidence during TPT. By maximizing the entropy of text feature dimensions, D-TPT regularizes feature distribution to reduce reliance on dominant dimensions. Experiments on multiple fine-grained and domain-shift datasets show that D-TPT achieves **better calibration and stable accuracy** than prior TPT variants, though its **novelty and theoretical depth are limited**.

**Strengths:**

1. **Motivation is Clear**
   The paper presents a clear motivation and introduces a novel perspective for analyzing model calibration. It reveals a new source of the modality gap — the dominant dimension — and proposes Dimensional Entropy Maximization (DEM) to suppress the excessive influence of these highly sensitive dimensions.

2. **Presentation is Good**
   The paper is well-written and logically structured. Starting from the modality gap problem, the authors identify the dominant dimension issue and further propose dimensional entropy maximization. The figures are particularly well-designed — for example, Figure 2 illustrates the discovered phenomenon, and Figure 5 clearly compares the proposed method with prior approaches. I like the figures in this paper.

3. **Experiments are Comprehensive**
   The experiments are extensive and demonstrate the stability and generality of the proposed method. The setup covers:
   * 11 fine-grained classification datasets and 4 out-of-distribution test sets (ImageNet-A/R/V2/Sketch)
   * Two backbone architectures (CLIP-ViT-B/16 and CLIP-RN50)
   * Five evaluation metrics (Accuracy, ECE, AECE, MCE, AURC)
   * Additional analyses including average and variance reporting, failure-case analysis, Pareto front analysis, and prompt initialization studies.

**Weaknesses:**

1. **Limited Novelty and Theoretical Depth**
   Although the authors claim the proposed method is effective, its novelty is limited, and the theoretical depth is weak. The core idea of D-TPT is highly similar to that of C-TPT and O-TPT — all add feature regularization on top of the TPT framework. D-TPT merely shifts from inter-feature to intra-feature regularization, still using a KLD + λ weighted form, without introducing a new learning mechanism or architecture.
   While the intuition behind Dimensional Entropy Maximization (DEM) is clear — reducing dominant dimension sensitivity by increasing feature entropy — the theoretical analysis remains mostly empirical.
   * Section 4.4 relies on geometric interpretation without mathematical derivation or quantitative validation.
   * Equation (3) defines the KLD loss against a uniform distribution, but it is unclear why this is theoretically equivalent to reducing dominant dimension sensitivity.
   * No theorem, proposition, or proof is provided to establish a causal link between DEM and calibration error.
   Overall, the main idea in this paper is more like an empirical heuristic regularization trick.

**Questions:**

1. I am still not sure why this method performs better than TPT. Unsupervised test-time prompt tuning fundamentally relies on confidence estimation of test samples, and TPT improves accuracy by amplifying model confidence. By contrast, removing dominant text or image features seems to intentionally suppress feature saliency, which should negatively affect model performance. How does D-TPT overcome this potential drawback and even better than the baseline methods on the Pareto front analysis?

2. In Figure 6, what are the underlying reasons for some failure cases? In Section 5.3, Why does D-TPT perform worse on the CLIP-RN50 backbone?

3. Some implementation details remain unclear. For instance, in Table 1, are the reported results averaged over 11 datasets? Given the large domain differences among them, is the proposed method effective across all datasets? Is the method dataset-agnostic — can it be applied to any image classification dataset?

---

> ### Author Response · Authors · 2025-11-20
>
> We appreciate the reviewer’s insightful comments. We address all the concerns and provide new experiments to support our contributions.
>
> ### **W1. Novelty and Theoretical Depth**
> In order to retain the practicality required for test-time adaptation, we deliberately aimed to achieve calibration improvements within a single loss function, avoiding multi-term objectives with multiple hyperparameters or additional modules that increase computational complexity. The reason for applying the same hyperparameters across all datasets and architectures is consistent with this fundamental motivation. Improving calibration via refined regularization remains a promising direction of research, and in this regard, we strongly believe that introducing a novel regularization term and offering a perspective that goes beyond diversity-based approaches constitutes a meaningful contribution rather than a minor modification. We agree that our proposal is grounded in empirical findings rather than theoretical claims. Even so, by elucidating the geometric behavior of prior methods and showing that a simple, modality-aware solution can improve calibration, we hope to provide an initial conceptual step toward connecting empirical observations with theoretical perspectives.
>
> ### **Q1. The reason why D-TPT outperforms TPT on some datasets**
> D-TPT does not surpass TPT across all datasets; however, on some datasets (e.g., Flower102), it shows better performance. One possible interpretation is that partial mitigation of the modality gap may facilitate improved performance [1]. To examine this hypothesis, we report results obtained by incorporating a loss term (GAP loss) that minimizes the angle between the text-feature centroid and the image features. The table below shows that applying the GAP loss enhances performance on Flower102, and that its combination with TPT leads to further improvements. While the GAP loss explicitly encourages alignment of text-feature dimensions with the image-feature distribution, D-TPT alleviates the modality gap indirectly by promoting a more uniform distribution across text-feature dimensions.
>
> |Method|Zeroshot|TPT|GAP|TPT w/ GAP|D-TPT|
> |-|-|-|-|-|-|
> |Acc. (%)|67.40|68.98|68.29|70.24|71.78|
>
> ### **Q2. About failure cases**
> Based on our geometric interpretation, D-TPT imposes stronger interference with the TPT objective for non–top-1 text features compared to C-TPT. This suggests that our method is particularly effective when TPT suffers from severe overconfidence, whereas C-TPT tends to be more effective in the opposite regime. Figure 6 illustrates a failure case in which the proposed method exhibits a comparatively higher ECE when TPT attains a well-calibrated state. The table below presents C-TPT and D-TPT performance over different ECE intervals of TPT, and the superiority of D-TPT becomes most pronounced when ECE of TPT is high. The same interpretation applies to the ResNet results; although CLIP-RN begins with an ECE that is 1% higher than ViT, applying TPT lowers its ECE by 0.25%. Overall, the proposed method is particularly effective in high-risk settings where TPT tends to amplify overconfidence.
>
> |ECE (%)|TPT|C-TPT|D-TPT|
> |-|-|-|-|
> |ECE$_{TPT}\>10%$|15.09|6.32|4.49|
> |ECE$_{TPT}\le10%$|5.00|2.63|3.63|
>
> ### **Q3. Generalization capability across diverse datasets**
> As the reviewer pointed out, Table 1 summarizes the average performance over the 11 datasets while the detailed results for each individual dataset are provided in Appendix A. The table below presents the experimental results on medical domain datasets. The results show competitive performance in the medical domain, indicating that our method can be effectively applied across diverse domains.
>
>
> >Baseline : MedCLIP, dataset: COVID
> |Clean| BAPLe|w/ C-TPT|w/ O-TPT|w/ D-TPT|Backdoor|BAPLe|w/ C-TPT|w/ O-TPT|w/ D-TPT|
> |-|-|-|-|-|-|-|-|-|-|
> |Acc. (%)|81.65|82.45|82.30|81.80|Acc (%)|99.30|99.95|99.95|99.75|
> |ECE (%)|20.33|22.91|15.60|12.85|ECE (%)|20.99 |26.01|5.70|4.79|
>
> >Baseline : MedCLIP, dataset: RSNA18
> |Clean| BAPLe|w/ C-TPT|w/ O-TPT|w/ D-TPT|Backdoor|BAPLe|w/ C-TPT|w/ O-TPT|w/ D-TPT|
> |-|-|-|-|-|-|-|-|-|-|
> |Acc. (%)|60.10|47.23|60.70|60.77|Acc (%)|98.83|99.97|99.43|99.40|
> |ECE (%)|15.74|18.53|15.28| 17.09|ECE (%)|10.20|13.45|11.65|11.45|
>
> >Baseline: PLIP, dataset: KatherColon
> |Clean| BAPLe|w/ C-TPT|w/ O-TPT|w/ D-TPT|Backdoor|BAPLe|w/ C-TPT|w/ O-TPT|w/ D-TPT|
> |-|-|-|-|-|-|-|-|-|-|
> |Acc. (%)|88.80|88.79|88.82| 88.80|Acc (%)|98.28|98.27|98.32|98.29|
> |ECE (%)|2.54|2.53|2.56|2.51|ECE (%)|1.46|1.46|1.46|1.50|
>
> ### **Reference**
> [1] Ouali, Yassine, et al. "Black box few-shot adaptation for vision-language models.", ICCV 2023.

---

> > ### Comment · Reviewer_oN3E · 2025-11-26
> > **I will keep my score unchanged.**
> >
> > Thanks for the reply. I still have concerns about the novelty part and the underlying causal reasons why D-TPT outperforms TPT. I will keep my score unchanged.

---

> ### Author Response · Authors · 2025-11-27
> **Official Response by Authors (1/2)**
>
> We appreciate the reviewer for carefully reading our comments and providing a fast response. We would like to provide additional clarification regarding the reviewer’s concerns.
>
> ## **1. Additional explanation on novelty**
>
> As detailed in our response to Reviewer zkoL (w1.innovation), the novelty of our work can be summarized as follows.
> (1) We aim to understand why calibration improves in multimodal settings by analyzing the modality gap phenomenon, which highlights the conceptual discrepancy between single-modal and multimodal calibration methods.
> (2) We identify dominant dimensions in both modalities and, based on this observation, propose dimensional entropy maximization as the simple and intuitive solution. To the best of our knowledge, this is the first work that *(i)* interprets diversity-based calibration methods through the lens of the modality gap, and *(ii)* achieves calibration by dimensional entropy maximization.
>
> To clarify how our proposed D-TPT differs from prior diversity-based methods such as C-TPT and O-TPT, we summarize the comparison in the table below. In contrast to C-TPT and O-TPT, which apply **regularization across classes**, D-TPT focuses on the **internal dimensional structure of each individual text feature**.
>
> |Method|C-TPT|O-TPT|D-TPT|
> |-|:-:|:-:|:-:|
> |Motivation|diversity|diversity|modality gap|
> |Problem statement|inter-class dispersion|inter-class angular separation|intra-class dominant dimension|
> |Objective|inter-class dispersion $\uparrow$|inter-class orthogonality $\uparrow$|intra-class dimensional entropy $\uparrow$|
>
> ## **2. Additional explanation regarding weaknesses**
> ### 2.1. Quantitative validation of Section 4.4
> **The numerical values shown on the hypersphere in Figure 5 of Section 4.4 correspond to the average logit range, logit mean, and ATFD values computed on the DTD dataset.** Since the logits are proportional to the cosine similarity between image and text features, an increase in the logit range indicates a larger difference between the maximum and minimum angles of text features relative to the image feature. Similarly, an increase in the logit mean reflects that the average angle between image and text features becomes smaller.
>
> Specifically, as shown in Figure 5 above (and summarized in the table below), the logit ranges are 10.92 (TPT), 10.88 (C-TPT), and 9.49 (D-TPT). The smallest logit range of D-TPT implies that the maximum–minimum angular spread of text features around the image feature is the smallest. Additionally, the logit mean values are 22.20 (TPT), 22.71 (C-TPT), and 23.23 (D-TPT), with D-TPT achieving the highest value, indicating that the average text feature is closer to the corresponding image feature.
>
> For ATFD, the values are 0.407 (TPT), 0.466 (C-TPT), and 0.409 (D-TPT). Although D-TPT exhibits a similar level of text dispersion to TPT, its smaller logit range demonstrates that D-TPT achieves diversity primarily along the binormal direction, rather than expanding excessively along the tangent direction. Our geometric illustration is directly derived from these empirical measurements.
>
> >The logit range, mean, and ATFD values are computed using the DTD dataset and correspond to the measurements reported in Figure 5 of the main paper.
>
> |Method|TPT|C-TPT|D-TPT|
> |-|:-:|:-:|:-:|
> |Logit Range|10.92|10.88|9.49|
> |Logit Mean|22.20|22.91|23.23|
> |ATFD|0.407|0.466|0.409|
>
> ### 2.2. Our objective function
> The reason we adopt a KLD loss rather than directly regularizing dominant dimensions or sensitivities is that converting features into probability distributions allows us to maintain stable magnitudes across different datasets and classes. To clearly demonstrate how KLD loss affects sensitivity, we compare the ratio of sensitivities between dominant dimensions and the remaining non-dominant dimensions, as shown in the table below. The dominant dimension sensitivity ratios are 63.86% (Zero-shot), 68.83% (TPT), and 63.64% (D-TPT). These results empirically show that while TPT increases sensitivity toward the dominant dimensions, D-TPT preserves the zero-shot sensitivity ratio.
>
> |Sensitivity|Dominant Dimensions|Non-Dominant Dimensions|
> |-|:-:|:-:|
> |Zero-shot|63.86%|36.14%|
> |TPT|68.83%|31.17%|
> |D-TPT|63.64%|36.36%|

---

> ### Author Response · Authors · 2025-11-27
> **Official Response by Authors (2/2)**
>
> ## **3. Case study on the performance improvement of D-TPT**
>
> To better understand the performance gains of D-TPT over TPT on the Flowers102 dataset, we analyzed the classes that showed meaningful improvements. In the Zero-shot CLIP, the cosine similarity between the text embeddings of “marigold” and “english marigold” is very high (0.9521), which results in poor separability and causes TPT to achieve only 15% recall for “marigold.” In contrast, D-TPT reaches 95% recall, representing an 80% improvement. Similarly, for “english marigold,” TPT achieves 45% recall, while D-TPT improves this to 55%.
>
> Since text features are normalized to unit length, reducing the influence of the dominant dimension naturally elevates the relative importance of the remaining dimensions. This observation is consistent with our sensitivity analysis. This enhances the contribution of non-dominant dimensions and allows the model to utilize a broader set of textual characteristics. These effects provide an explanation for the improved performance of D-TPT compared to TPT on the Flowers102 dataset.
>
> |Recall|TPT|D-TPT|
> |-|:-:|:-:|
> |“marigold”|15%|95%|
> |“english marigold”|45%|55%|
>
> ---
>
> *We hope that our additional explanations help address the reviewer’s concerns.*

---

### Official Review · Reviewer_zkoL · 2025-10-30

**Soundness:** 2
**Presentation:** 2
**Contribution:** 2
**Rating:** 4
**Confidence:** 4

**Summary:**

This work is driven by the observation that text and video feature vectors have non-aligned dominant features.  The method therefore seeks to reduce the calibration error of TPT by regularizing the feature vectors so that they are moved toward the all-zero vector.  As expected, this proposal reduces the overconfidence effect of TPT, but it also usually reduces accuracy.

The idea feels a little bit trivial, in the sense that there are many regularization algorithms that push many types of weight vectors and feature vectors toward the all-zero vector.  On the other hand, it has not previously been observed that pushing the prompt text embedding toward the all-zero vector improves calibration, or that the effectiveness of this step is well-motivated by the observation that each text vector is strongly dominated by one of its dimensions.  On balance, it feels like this observation needs a little bit more rigorous analysis before publication.

I'm also concerned about the apparent errors in the presentation of TPT in Eq. (1) and Algorithm 1.

**Strengths:**

* The connection between dominant features and zero-directed regularization has not been previously applied to prompt tuning, as far as I know.
* Zero-directed regularization of test-time prompts is easy to implement, and if the experimental results hold up, could be quite useful.

**Weaknesses:**

Motivation/Innovation:

This proposal reduces the distance between an image vector and its corresponding text vector, as claimed, but only because it is shifting ALL feature vectors in the direction of the all-zeros vector (the vector whose sigmoid transformation has the lowest KL divergence to a uniform distribution).  It is well known that regularizing feature vectors will reduce overconfidence: the only new contribution of this paper is to point out that regularization of this kind also works for TPT.

Since this method also reduces accuracy relative to unmodified TPT, it's not clear that it is beneficial.

The motivation for this proposal is the observation, in Figure 2 and Figure 3(a), that the text and image feature vectors are each dominated by one dimension, and that the dominant dimension differs by modality.  Figures 2 and 3(a) only demonstrate this effect for two individual cases, however. The text claims that this example is typical, but provides no proof.  Actually other papers have also reported this effect, but I've never seen any quantification of the size of this effect.

Correctness:

Eq. (1) is an incorrect statement of TPT, and the part of Algorithm 1 that is claimed to reimplement TPT does not do so correctly.  The TPT objective is the class entropy of an averaged probability; the averaged probability is computed as the average across high-confidence augmented images.  Eq. (1) and Algorithm 1 perform averaging of the entropy across images, rather than averaging the probability.  This could have strange effects in some cases, e.g., the proposed incorrect formula might incorrectly select a prompt that causes different augmentated images to confidently predict different answers, in preference to a prompt that causes each augmentation to choose the same answer but with lower confidence.

The averaging should only be across augmented views of the same image; the text before Eq. (1) suggests that i=1 to N includes multiple images, not just multiple augmentations, which would be another error in this supposed reimplementation of TPT.

Results:

Table 3(b) shows that average accuracy degrades when the dominant dimension is replaced by its average, contrary to claims in the text.

Results in Tables 1 and 2 have the proposed algorithm highlighted in all columns, even though it is not the best in all columns.  In particular, TPT usually has better accuracy, O-TPT usually has better AURC, and proposed algorithm usually has the best ECE and AECE.

English usage:

There are a number of small English usage errors, e.g., p. 1 par. 1: Based on the observation... --- This sentence lacks a verb.

**Questions:**

See "Weaknesses."

---

> ### Author Response · Authors · 2025-11-20
>
> We thank the reviewer for the constructive feedback. We address the concerns and provide detailed descriptions of our contributions.
>
> ### **W1. Innovation**
> Research aiming to improve calibration capability by regularizing logit norms achieves successful results in single-modal settings [1], but observations suggest that logit range is more important than logit magnitude in multimodal settings [2]. Our proposed method begins by considering the differences between single-modal and multimodal approaches to calibration. If text embeddings are interpreted as counterparts to classifier weights in single-modal networks, they naturally exhibit uniformity as a consequence of Neural Collapse [3]. By contrast, multimodal VLMs such as CLIP do not exhibit this behavior, which leads us to hypothesize that improving the uniformity of text embeddings may enhance calibration, reflecting intrinsic properties of contrastive VLMs. We believe that our contribution is attractive in offering an explanation for the success of diversity-based methods through the modality-gap phenomenon in CLIP, where image and text embeddings reside in distinct local subspaces. Furthermore, we aim to present additional possibilities for exploring the calibration problem, which has thus far been studied largely through diversity-based approaches. Moreover, since CLIP’s logit magnitude is directly proportional to the cosine similarity between image and text features, zero-directed regularization reduces cosine similarity and thereby enlarges the modality gap. To clarify the distinction from naive zero-directed regularization, we present experiments that apply a direct penalty to the logit norm.
>
> |Method|Metric|DTD|Flw|Food|SUN|Air|Pets|Cal|UCF|Euro|Cars|Mean|
> |---|---|---|---|---|---|---|---|---|---|---|---|---|
> |Logit Norm|Acc. (%)|44.21|67.84|83.4|61.88|21.81|87.95|92.13|62.89|43.01 |63.56|62.87|
> |   | ECE (%)|7.55|3.49|1.78|2.99|10.49|2.56|5.13|5.17|7.25|2.03|4.84|
> | D-TPT | Acc. (%) |45.09|71.78|83.10|64.46|24.06|88.25|93.14|66.67|44.09  |64.21|64.49|
> |   | ECE (%)|5.21|2.33|4.05|4.33|3.77|2.36|5.49|2.53|9.68| 2.27|4.20|
>
> ### **W2. Motivation**
> We include graphs for each dimension across various datasets and backbone architectures in Appendix C. Specifically, Figure 8 of the Appendix C shows that a dominant dimension consistently appears in CLIP-ViT/16 and CLIP-RN across ImageNet, ImageNet-A, ImageNet-K, FGVC-Aircraft, and StanfordCars. Additionally, Figure 9 of the Appendix C presents experimental results on sensitivity. While some variations exist across datasets, the results confirm that the dominant dimension of the text embedding consistently exhibits high sensitivity.
>
> ### **W3. Correctness**
> Thank you for pointing out the incorrect formula. We corrected the wrong formula for TPT in Eq (1) in the main text, and you can find it in the revised version. For our implementation, we followed the official TPT code as provided in the supplementary material, and all experimental results were conducted using the marginal entropy minimization method corresponding to the corrected formula. Averaging is also applied using $N−1$ augmentations of a single image, and thus the resulting entropy reflects that of a single image rather than multiple images. For clarity, we have updated line 154 in the main paper.
>
> ### **W4. Results**
> Our claim regarding Table 3(b) in the main paper is not that averaging over the dominant dimension always improves accuracy, but that it leads to an improvement in the average ECE. In addition, as described in line 206, we observe cases in which both accuracy and ECE improve, specifically for the DTD and Caltech datasets.
> The background color highlighting in Tables 1 and 2 is intended to improve visibility as an indicator of the proposed method, rather than to emphasize the best performance. As the reviewer pointed out, our method does not achieve the best results across all metrics. Although ECE is a standard metric for calibration and our method performs well under this metric, no single metric fully characterizes the calibration properties of a model [4]. For this reason, we aimed to report the results transparently by including additional evaluation metrics such as AECE, MCE, and AURC.
>
> ### **W5. Minor revisions**
> Thank you for carefully reviewing our paper. The grammatical inaccuracies and typos have been corrected.
>
> ### **Reference**
> [1] Wei, Hongxin, et al. "Mitigating neural network overconfidence with logit normalization." ICML 2022.
> [2] Murugesan, Balamurali, et al. "Robust calibration of large vision-language adapters." ECCV 2024.
> [3] Papyan, Vardan, et al. "Prevalence of neural collapse during the terminal phase of deep learning training.", PNAS, 2020.
> [4] Pavlovic, Maja. "Understanding Model Calibration-A gentle introduction and visual exploration of calibration and the expected calibration error (ECE).", The Fourth Blogpost Track at ICLR 2025.

---

### Official Review · Reviewer_HJQF · 2025-11-01

**Soundness:** 3
**Presentation:** 3
**Contribution:** 3
**Rating:** 6
**Confidence:** 4

**Summary:**

The paper studies why CLIP-based test-time prompt tuning (TPT) tends to improve accuracy but hurt calibration, and it argues that the root cause is not just lack of inter-class feature dispersion (as in C-TPT, O-TPT) but an overreliance on a few dominant dimensions in the text/image embeddings that create a modality gap. To address this, the authors propose D-TPT, which keeps the standard TPT entropy-minimization loss but adds a dimensional entropy maximization term that pushes each text feature to distribute its mass more uniformly across embedding dimensions, thereby reducing the influence of the dominant dimension. Across 11 fine-grained benchmarks and 4 ImageNet-shift datasets, D-TPT largely preserves the accuracy gains of TPT while recovering or improving calibration (ECE, MCE) compared to existing TPT variants.

**Strengths:**

* Calibration in CLIP test-time prompt tuning (TPT) has been studied quite a bit lately, with methods like C-TPT [1], O-TPT [2], and A-TPT [3]. Most of these approaches build on the same core intuition first introduced in C-TPT, that improving feature dispersion helps calibration. This paper takes a different angle: instead of proposing yet another dispersion-based variant, it digs into why TPT on CLIP becomes miscalibrated in the first place and points to dominant dimensions / modality gap as a causal factor. This seems like a meaningful technical contribution to the field.

[1] https://arxiv.org/abs/2403.14119

[2] https://arxiv.org/abs/2503.12096

[3] https://www.arxiv.org/abs/2510.26441

* The paper reports multiple calibration error metrics other than ECE, which strengthens the empirical evaluation and makes the conclusions about calibration more reliable.

**Weaknesses:**

* It would be nice to see if such method works on critical domains such as medical domain.

* The results show that suppressing the dominant dimension can help, and then infer it is the main driver. But they don’t fully rule out alternative explanations (e.g. regularization just reduces logit range in general). So the 'causal factor' framing is a bit stronger than what the experiments actually prove.

* Since the proposed D-TPT regularizes intra-feature dimensional entropy, whereas prior methods such as C-TPT and O-TPT focus on inter-feature dispersion/orthogonality, it would be valuable to examine whether the two types of regularization are complementary. For example, can D-TPT be applied on top of C-TPT’s feature dispersion term or O-TPT’s orthogonality constraint.

**Questions:**

See weaknesses above.

---

> ### Author Response · Authors · 2025-11-20
>
> We appreciate the reviewer’s insightful comments. We address all the concerns and provide new experiments to support our contributions.
>
> ### **W1. Experimental results on medical domains**
> We appreciate the reviewer’s suggestion to evaluate our method on medical domain tasks, where calibration is especially important. We report experimental results obtained by applying C-TPT, O-TPT, and D-TPT to BAPLe [1], as presented in the table below and further detailed in Appendix B. During training, the loss function corresponding to each method is incorporated into the objective function of BAPLe. Following BAPLe, we use PLIP [2] as the base model for Kather, whereas MedCLIP [3] is used for the other datasets. The results show competitive performance in the medical domain, indicating that our method can be effectively applied across diverse domains.
>
> >Baseline : MedCLIP, dataset: COVID
> |Clean| BAPLe|w/ C-TPT|w/ O-TPT|w/ D-TPT|Backdoor|BAPLe|w/ C-TPT|w/ O-TPT|w/ D-TPT|
> |-|-|-|-|-|-|-|-|-|-|
> |Acc (%)|81.65|82.45|82.30|81.80|Acc (%)|99.30|99.95|99.95|99.75|
> |ECE (%)|20.33|22.91|15.60|12.85|ECE (%)|20.99 |26.01|5.70|4.79|
>
> >Baseline : MedCLIP, dataset: RSNA18
> |Clean| BAPLe|w/ C-TPT|w/ O-TPT|w/ D-TPT|Backdoor|BAPLe|w/ C-TPT|w/ O-TPT|w/ D-TPT|
> |-|-|-|-|-|-|-|-|-|-|
> |Acc (%)|60.10|47.23|60.70|60.77|Acc (%)|98.83|99.97|99.43|99.40|
> |ECE (%)|15.74|18.53|15.28| 17.09|ECE (%)|10.20|13.45|11.65|11.45|
>
> >Baseline: PLIP, dataset: KatherColon
> |Clean| BAPLe|w/ C-TPT|w/ O-TPT|w/ D-TPT|Backdoor|BAPLe|w/ C-TPT|w/ O-TPT|w/ D-TPT|
> |-|-|-|-|-|-|-|-|-|-|
> |Acc (%)|88.80|88.79|88.82| 88.80|Acc (%)|98.28|98.27|98.32|98.29|
> |ECE (%)|2.54|2.53|2.56|2.51|ECE (%)|1.46|1.46|1.46|1.50|
>
> ### **W2. Causal factor**
> Our concern is that existing attempts to address the calibration in test-time prompt tuning have relied on diversity-based approaches. We aim to extend beyond this direction by introducing a new perspective grounded in the modality characteristics of CLIP. In this regard, our claim is not that the proposed method identifies a fundamental causal factor; rather, it shows that calibration capability can also be achieved by reducing diversity, specifically by alleviating the influence of the dominant dimension. This observation suggests that the fundamental causal factor may lie beyond diversity or the dominant dimension itself, and that there may exist deeper causal factors or hidden properties worth exploring. In this study, we attempt to interpret both the overconfidence of TPT and the success of diversity-based methods through the lens of the modality gap. Although this interpretation is not theoretically proven and remains a conjecture, our results demonstrate that calibration can be enhanced even with a simple yet effective method that increases entropy across dimensions.
>
> ### **W3. Combination of C-TPT and D-TPT**
> We report the results of combining C-TPT and D-TPT in the table below. Each loss function is added to the TPT objective without any hyperparameter adjustment. Although incorporating C-TPT into D-TPT improves ECE for DTD, Flowers, Aircraft, and StanfordCars, we also observe cases with marginal or degradation, and accuracy consistently decreases. This indicates that naive combinations may induce strong regularization, which can be detrimental to performance. Moreover, identifying appropriate balancing parameters for each method would require access to training or validation data, which is incompatible with the constraints of the TTA setting and undermines practical usability. Nonetheless, we believe that D-TPT could be integrated with diversity-based methods through carefully balanced hyperparameters or selective regularization. Motivated by failure cases analysis, we additionally report results for D-TPTxC-TPT, which selectively applies C-TPT or D-TPT based on entropy. D-TPTxC-TPT yields improved mean ECE.
>
> |Combination|Metric|DTD|Flw|Food|SUN|Air|Pets|Cal|UCF|Euro|Cars|Mean|
> |-|-|-|-|-|-|-|-|-|-|-|-|-|
> |C-TPT| Acc. (%)|45.51|69.22|83.00|64.31|24.09|88.44|93.55|64.90|42.32| 65.70|64.10|
> | |ECE (%)| 12.54|5.48|3.25|4.89|4.29|1.65|3.96|2.85|11.49|1.19|5.16|
> |D-TPT| Acc. (%)|45.09|71.78|83.10|64.46|24.06|88.25|93.14|66.67|44.09| 64.21|64.49|
> | |ECE (%)| 5.21|2.33|4.05|4.33|3.77|2.36|5.49|2.53|9.68|2.27|4.20|
> |D-TPT+C-TPT|Acc (%)|44.68|71.34|83.09|64.13|23.64|88.23|92.82|65.29|40.14|64.32|63.77|
> | |ECE (%)|4.84|1.85|4.04|5.14|3.63|2.51|6.02|2.67|12.08|2.12|4.49|
> |D-TPTxC-TPT|Acc. (%)|44.33|71.62|82.95|64.65|24.06|88.39|93.51 |66.11|40.05|64.68|64.44|
> | | ECE (%)|5.32|2.82|3.59|5.07|3.80|1.98|4.17|2.28|9.70|1.83|4.10|
>
> ### **Reference**
> [1] Hanif, Asif, et al. "Baple: Backdoor attacks on medical foundational models using prompt learning.", MICCAI 2024.
> [2] Huang, Zhi, et al. "A visual–language foundation model for pathology image analysis using medical twitter.", Nature medicine, 2023.
> [3] Wang, Zifeng, et al. "Medclip: Contrastive learning from unpaired medical images and text.", EMNLP 2022.

---

### Author Response · Authors · 2025-11-20
**General Response**

Thank you for providing constructive insights on our paper and for taking the time to review the paper. We have revised the manuscript as outlined below to incorporate reviews comments, with the revised manuscript written in **blue**.
1. We have added new experimental results for the medical domain in Appendix B.
2. We have added new results for dominant dimensions and sensitivity across various datasets and architectures in Appendix C.
3. We have corrected the formula for marginal entropy and fixed minor typos for better readability.

Additional discussions will be incorporated in a subsequent revision.

---

### Author Response · Authors · 2025-12-03
**Summary of discussion period**

We sincerely thank the reviewers for providing constructive feedback and would like to express our gratitude to the reviewers, ACs, SACs, and PCs for their dedication and service to our community.

In this paper, we aim to understand why diversity-based calibration methods for VLMs are effective and to provide a new perspective on the modality gap. We identify the presence of dominant dimensions underlying the modality gap in VLMs and observe that mitigating the text-dominant dimensions that exhibit high prediction sensitivity can help improve calibration. Motivated by these observations, our contribution lies in proposing dimensional entropy maximization as a simple yet effective solution and in interpreting existing methods within the geometric structure of CLIP through the lens of the modality gap.

The reviewers acknowledged our work for offering a new perspective beyond prior diversity-based interpretations, conducting extensive experiments, presenting a clear and logical structure. Following reviewers constructive feedback, we can have a chance to revise the manuscript to clarify our contributions. Additionally, we believe that this constructive process of discussion and revision will help dispel the reviewers’ concerns.

- **Improved clarity of novelty.** We improved the clarity of our novelty by adding a description in the Introduction (Section 1) that motivates our work by highlighting the differing calibration behaviors between single-modal models and VLMs.

- **Correction of equation.** We corrected the marginal entropy loss of TPT in Equation (1) to reflect the proper formulation.

- **Discussion on causal chains.** We clarified the causal chain from the dominant dimension to overconfidence in Geometric Support (Section 4.4) by providing the subsection “From dominant dimension to overconfidence” and connecting the discussion more explicitly to the accompanying quantitative evidence for better understanding

- **Additional analysis of failure cases.** We improved the clarity of the analysis by incorporating an additional comparison based on performance across TPT’s ECE intervals for the method’s success and failure cases in Figure 7 (Section 5.3).

- **Additional visualization of dominant dimensions and sensitivity.** We provided additional visualizations in Figures 9 and 10 (Appendix B) to show that the modality gap phenomenon and the sensitivity associated with dominant dimensions broadly appear across a wide range of architectures and datasets.

- **Domain extension to demonstrate generality.** We supported the generality of our method by evaluating it in a calibration critical medical domain and reporting the comparison with existing methods in Table 8 (Appendix C).


Once again, we would like to express our sincere gratitude to reviewers, ACs, SACs, and PCs.
Sincerely,
Authors of Paper #8653

---

### Meta-Review · Area_Chair_Btvp · 2025-12-31

**Summary:**

The reviewers' opinions overall tend towards rejection. Three reviewers gave a score of 4 "Borderline Reject", and one gave a 6 "Marginally Accept". While the reviewers acknowledged the extensive experimental evaluation in the paper and the attempt to consider calibration from the perspective of modality gap, several key issues led to the current decision:

Limited novelty: A major shared criticism is that the proposed D-TPT method is a minor improvement over existing methods such as C-TPT and O-TPT. Reviewers felt that shifting from inter-class to intra-class regularization represents a heuristic adjustment rather than a fundamental methodological breakthrough. The method was described by some as essentially regularizing features toward a zero vector or uniform distribution, which is a well-known technique with limited novelty in this context.

The theoretical section is not convincing: The paper’s core theoretical claim (a "dominant dimension" leads to the modality gap, which in turn causes overconfidence) did not persuade the reviewers. Although the authors attempted to explain this in their response, the reviewers still believe this causal chain lacks formal mathematical proof and remains largely conjectural. The geometric explanations provided in the paper are insufficient to establish a solid theoretical link between dimensional entropy and calibration error.

Trade-off between accuracy and calibration: There are significant concerns about the stability of the method’s performance. The reviewers noted that while D-TPT improves calibration metrics (such as ECE), it often reduces classification accuracy (for example, on the CLIP-RN50 backbone). This suggests that the method actually suppresses feature saliency to reduce confidence, which may weaken the model’s discriminative ability.

**Reviewer Concerns:**

Concerns Addressed by the Rebuttal:
1. Experimental Scope: The authors successfully addressed Reviewer HJQF's request for evaluation on critical domains by providing new results on medical datasets (MedCLIP, PLIP, etc.).
2. Technical Correctness of TPT Formulation: The authors acknowledged and corrected the error in Equation (1) regarding the TPT objective function, which was flagged by Reviewers zkoL and 5wvu. They clarified that their implementation code was correct despite the textual error.
3. Interaction with Prior Methods: The authors provided additional data on combining D-TPT with C-TPT as requested by Reviewer HJQF, clarifying that naive combinations can degrade performance due to excessive regularization.
4. Clarifications on Notation and Baselines: The authors clarified the notation regarding normalized text features (Reviewer 5wvu) and provided comparisons to simple logit normalization to differentiate their method (addressing Reviewer zkoL).

Outstanding Concerns:
1. Limited Novelty: Reviewers oN3E and 5wvu remain concerned that the proposed method lacks significant innovation. They view D-TPT as a minor variation of existing regularization techniques or a heuristic trick, rather than a substantial methodological advancement. Reviewer oN3E explicitly stated that the novelty concerns were not alleviated by the rebuttal.
2. Theoretical Grounding and Causal Links: A major outstanding issue, particularly for Reviewer 5wvu, is the lack of a rigorous proof or convincing explanation for the causal chain: "dominant dimension — modality gap — overconfidence." Despite the authors' additional explanations and geometric interpretations, the reviewers found the link to be largely empirical and conjectural rather than theoretically sound.
3. Performance Stability and Trade-offs: Reviewers zkoL and oN3E noted that while calibration improves, accuracy often degrades (e.g., on CLIP-RN50). The concern that suppressing dominant dimensions intentionally reduces feature saliency—thereby hurting model discriminability—remains a valid criticism that was not fully resolved by the provided failure case analysis.

**Reviewer Scores:**

Based on the reviewers' initial comments, the authors' rebuttal, and the subsequent discussion (or lack thereof), here is the assessment of the scores:

Reviewer HJQF (Current Score: 6)
I believe this reviewer would have kept score at 6. The authors directly addressed this reviewer’s main request by adding experiments in medical domains (MedCLIP, PLIP), which showed competitive performance. Had this reviewer joined the discussion, he would likely have been satisfied with the new data. However, the theoretical concerns raised by other reviewers would probably have kept the score at 6.

Reviewer zkoL (Current Score: 4)
I believe this reviewer would have maintained their score of 4. Their fundamental criticism was that the method feels "trivial" and that it degrades accuracy. While the authors corrected the mathematical error in the TPT formulation that zkoL pointed out, the rebuttal regarding the "novelty" of the regularization likely would not have been sufficient to overcome the reviewer's view. This reviewer believe the proposed method is a standard heuristic with limited innovation.

Reviewer oN3E (Current Score: 4)
This reviewer took part in the rebuttal discussion and explicitly stated would keep score unchanged at 4. Even after the authors clarified the difference between intra-class and inter-class regularization, the reviewer remained unconvinced about the method’s novelty and the underlying reasons for its performance.

Reviewer 5wvu (Current Score: 4)
This reviewer participated in the rebuttal discussion and clearly indicated would maintain score at 4. This reviewer specifically noted that the “causal chain” connecting dominant dimensions to overconfidence remains unresolved and lacks formal proof, and that the methodological novelty appears limited.

---

### Decision · Program_Chairs · 2026-01-26

Reject